


# Review article: Towards multi-hazard and multi-risk indicators – a review and recommendations for development and implementation

Christopher J. White[1]; Mohammed Sarfaraz Gani Adnan[2]; Marcello Arosio[3]; Stephanie Buller[4]; YoungHwa Cha[1]; Roxana Ciurean[5]; Julia M. Crummy[6]; Melanie Duncan[6]; Joel Gill[4]; Claire Kennedy[1]; Elisa Nobile[3]; Lara Smale[6] and Philip J. Ward[7,8]

[1] Department of Civil and Environmental Engineering, University of Strathclyde, Glasgow, G1 1XJ, UK
[2] Department of Civil and Environmental Engineering, Brunel University London, London, UB8 3PH, UK
[3] Department of Science, Technology and Society, Istituto Universitario di Studi Superiori di Pavia (IUSS), Pavia, 27100, Italy
[4] School of Earth and Environmental Sciences, Cardiff University, Cardiff, CF10 3AT, UK
[5] British Geological Survey, Nicker Hill, Keyworth, Nottingham, NG12 5GG, UK
[6] British Geological Survey, The Lyell Centre, Research Avenue South, Edinburgh, EH14 4AP, UK
[7] Department of Water and Climate Risk, VU Amsterdam, 1081 HV Amsterdam, The Netherlands
[8] Deltares, Delft, 2629 HV Delft, The Netherlands

*Correspondence to:* Dr Christopher J. White (chris.white@strath.ac.uk)

Department of Civil and Environmental Engineering, University of Strathclyde, James Weir Building, 75 Montrose Street, Glasgow, G1 1XJ, UK


**Abstract**
Undertaking a natural hazard or risk assessment from a single hazard approach can be considered incomplete
where the interactions between and impacts from multiple hazards and risks are not considered. However, the
development of indicators in disaster risk management has only recently started to explicitly include multi-hazard
and multi-risk approach. Indicators contain observable and measurable characteristics to simplify information to
understand the state of a concept or phenomenon, and/or to monitor it over time. To date, there have been limited
efforts to understand how indicators are being used in this context. Using a systematic review, 194 publications
were identified that mention indicators, covering hazards, vulnerability, and risk/impact. We find that the majority
of studies exploring indicators are multi-layer single hazards and risks; in other words, they did not include the
interactions between hazards. The results also demonstrate a predominance of studies on hazard indicators (88%)
versus risk indicators, with a dominance of hydro-meteorological indicators. Only 20% of the studies integrated
hazard, vulnerability and risk/impact. Based on the findings, we propose 12 recommendations to enable the uptake
of indicators, from advancing research into multi-hazard and multi-risk indicator frameworks, to enabling
partnerships to ensure the inclusion of stakeholder needs in indicator development.



## 1. Introduction

Natural hazard events have the potential to impact areas over diverse temporal and spatial scales as well as influence each other (Gill and Malamud, 2014). These events also impact environments where there may be overlapping dynamic vulnerabilities and exposure from the socio-economic conditions of affected areas (Johnson et al., 2016). Undertaking a natural hazard or risk assessment using a single hazard approach can be considered incomplete as these approaches do not consider the possible interactions and impacts from multiple hazards on a specific location (Gill and Malamud, 2016; Sekhri et al., 2020). Despite this, natural hazards and their associated risks have largely been investigated from a single hazard perspective. However, in recent years there has been an increased focus on both multi-hazard and multi-risk approaches (e.g., Kappes et al., 2012; Duncan et al., 2016; Ward et al., 2022). Here multi-hazards are defined as "(1) the selection of multiple major hazards that the country faces, and (2) the specific contexts where hazardous events may occur simultaneously, cascadingly, or cumulatively over time, and taking into account the potential interrelated effects" (UNDRR, 2017a).

The international shift from single to multi-hazard and multi-risk thinking began in the 1990s, initially with the United Nations Agenda 21 where pre-disaster planning and settlement planning recommended the inclusion of "complete multi-hazard research into risk and vulnerability" (United Nations, 1992). This was followed by the specification of "an integrated, multi-hazard, inclusive approach to address vulnerability, risk assessment and disaster management" (United Nations, 2002) from the World Summit on Sustainable Development. In 2005, the Hyogo Framework for Action – with the aim of reducing disaster losses by 2015 – was adopted at the World Conference on Disaster Reduction. This framework called for the implementation of a multi-hazard approach to disaster risk reduction (UNISDR, 2005) and its incorporation into policies and planning for sustainable development. The Sendai Framework for Action (successor to the Hyogo Framework) inspires a multi-hazard approach to disaster risk reduction (DRR) practices (United Nations, 2015).

Aligned with the development and expansion of international DRR approaches, many indicators have been introduced to help assess the level of risk, monitor progress, and guide policies and interventions aimed at reducing disaster risk. Indicators are "observable and measurable characteristics that can be used to simplify information to help understand the state of a concept or phenomenon, and/or to monitor it over time to show changes or progress towards achieving a specific change" (Gill et al., 2022 adapted from; Ivčević et al., 2019); see Box 1 (containing Fig. 1). They can be used as a standard, to assist with making decisions and for communications, and are capable of capturing a broad range of physical, social, and economic parameters. Indicators are used as a tool to define a baseline and track changes for monitoring and evaluation, allowing for the simplification of information, a situation, or an event, allowing them to be better understood, replicated, and monitored over time. Indicators have been used in a wide range of ways and applications, including as single variables representing an environmental or climatic parameter. For example, a precipitation indicator may be used to represent flood occurrence or as an indicator of a meteorological drought (AghaKouchak et al., 2023). Other studies use indices that integrate a combination of indicators to account for a relationship between them, such as the Multivariate Standardized Drought Index that uses a combination of precipitation and soil moisture (AghaKouchak et al., 2023).

**Box. 1: Indicators and the Sendai Framework for DRR: from single to multi-hazards**





The Sendai Framework for Disaster Risk Reduction (2015–2030) highlights the necessity of multi-hazard risk assessments and encourages countries to adopt indicators that account for the interactions between different hazards. One tool developed by the UNDRR to help cities assess their resilience to disasters in line with the goals of the Sendai Framework is the Disaster Resilience Scorecard for Cities (https://mcr2030.undrr.org). The Scorecard is based on the Framework's four key priorities and provides specific indicators for a range of assessment levels. There are 47 indicators used for the preliminary level and 117 indicator criteria for a detailed assessment. While the Scorecard highlights the importance of identifying and understanding how multiple hazards "might combine, and how repeated small scale disaster events might accumulate in their impact over time" (UNDRR, 2017b p.14), there are no clear metrics associated with interacting multi-hazards. Instead, the emphasis is on cascading impacts between city infrastructure systems under different scenarios.

A more recent initiative for achieving the goals outlined in the Sendai Framework (specifically, Target G) is Early Warnings for All (EW4All), launched in 2022 and co-led by the WMO and UNDRR. As of 2023, 101 countries reported having Multi-Hazard Early Warning Systems (MHEWS), double the number of countries reported in 2015 (UNDRR and WMO, 2023). Progress reporting is through a set of custom indicators that are divided into four areas: disaster risk knowledge; detection, monitoring, analysis and forecasting of the hazards and possible consequences; warning dissemination and communication; and preparedness and response capabilities. Indicators in each area are computed using different methodologies and data sources. Progress is measured using either a binary approach (where 1 = yes, or indicator met, and 0 = no, or indicator not met) or a scale between the two values, pending on the computation method (UNDRR and WMO (2022). One of these, Indicator 2.2 (see Fig. 1), measures if 'multiple hazards and cascading hazardous events are assessed and translated into

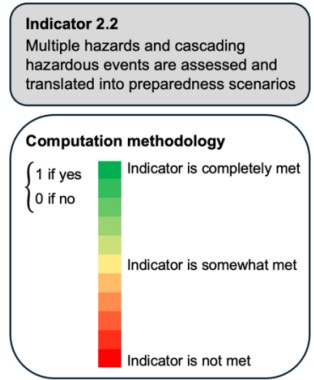

preparedness scenarios' using a binary approach. According to this methodology, the scoring should be validated using data sources such as: social, environmental, and physical vulnerability assessments; environmental management, response, and contingency plans; multi-hazard risk assessments or risk that consider effects from hazards that occur simultaneously, in cascade or cumulatively over time, and take into account the potential interrelated effects; or assessments considering climate change impacts. It is important to mention that the custom indicators focus on the minimum standard of risk knowledge required to make a MHEWS effective.

**Figure 1. An example of measuring MHEWS development using Indicator 2.2, measuring if 'multiple hazards and cascading hazardous events are assessed and translated into preparedness scenarios' (adapted after UNDRR and WMO (2022)). Progress is measured using a scale from 0 to 1, where 1 = indicator met (yes), and 0 = indicator not met (no).**


The International Decade for Natural Disaster Risk Reduction (IDNDR), which was declared by the United

Nations between 1990 and 1999 (United Nations Department of Humanitarian Affairs, 1994), saw the growth and



use of single hazard and single risk indicators. Today, the use of single hazard and single risk indicators are
commonplace (see Box 1). The development of multi-hazard and multi-risk indicators for disaster risk assessment
and management has not kept pace with the development of multi-hazard DRR approaches and the use of
indicators more generally. While indicator-based methods are commonly used to assess hazards and the
vulnerability of elements at risk, these approaches are limited as they do not integrate analyses of different hazards
or the interaction between them (Julià and Ferreira, 2021). Adopting a multi-hazard and multi-risk approach with
indicators would allow for the identification of interactions and the subsequent impacts of various hazards that
could be used to improve the understanding of both hazards and risk (Depietri et al., 2018). However, existing
approaches largely remain insufficient to support a multi-hazard analysis that take account of the complex
interactions between hazards (Lou et al., 2023a) and it remains a challenge to represent the dynamic nature of
hazards, exposure, vulnerability, and multiple risks. Cardona (2005) is one of the earlier works in this field,
presenting a framework for assessing and managing disaster risks by using indicators that account for various
hazards and vulnerabilities in Latin America and the Caribbean – a region particularly prone to several natural
hazards. However, the development of multi-hazard and multi-risk approaches was in its infancy at that time,
limiting the adoption and uptake of the concepts presented. More recent approaches advocating for the
development and use of multi-hazard and multi-risk indicators have been seen across a range of climate change
adaptation and disaster risk-related studies focusing on hazards, vulnerability, or exposure, but also impact, coping
capacity, and resilience. AghaKouchak et al. (2023) for example calls for drought monitoring and research to
"move beyond individual drivers and indicators to include the evaluation of various potential cascading hazards"
and to develop indicators that establish links between different hazards and the impact. In an assessment of coastal
resilience frameworks that also investigated the use of resilience indicators, Almutairi et al. (2020) note that most
of the frameworks evaluated consider single hazard types only, and that future frameworks should address the
interrelationships between multiple hazards. Sebesvari et al. (2016) similarly calls for a multi-hazard assessment
of vulnerability with the development of new indicators that would be able to capture the complexity and exposure
of multi-hazards, particularly in delta socio-ecological systems and regions. There remains, however, a gap in
knowledge as to what multi-hazard and multi-risk indicators have been developed or a clear demonstration of
what their potential is.
Terminology is a particular issue that has affected the development and uptake of multi-hazard indicators up to
now. For example, there are different ways of describing the interaction between hazards. These include triggering
or cascading relationships, where a primary hazard may cause an associated hazard; compound relationships,
where multivariate events and unrelated hazards may overlap spatially and/or temporally; and (de-)amplification,
where one decreases or increases the probability of occurrence or the magnitude of another hazard (Ciurean et al.,
2018; Gill et al., 2022). There are also alternative terms for what an indicator is, including index and metric. In
some instances, these terms are used interchangeably even though there is a distinction between their definitions,
i.e., an indicator is a single measurable variable or metric that provides information about a specific aspect of a
system, condition, or outcome; whereas an index is a composite measure that combines multiple indicators into a
single numerical value or score (OECD, 2008). To establish consistency, a set of definitions are provided in Table
128 1.

**Table 1. List of terms and definitions used in this study.**



| Terminology | Definitions | Source |
|---|---|---|
| Multi-hazard | "1) the selection of multiple major hazards that the country faces, and 2) the specific contexts where hazardous events may occur simultaneously, cascadingly or cumulatively over time, and taking into account the potential interrelated effects." | UNDRR (2017a) |
| Multi-risk | Risk generated from multiple hazards and the interrelationships between these hazards (and considering interrelationships on the vulnerability level). | Zschau (2017) |
| Compound hazards | "Compound weather and climate events are defined as a combination of multiple drivers and/or hazards that contribute to risk." | Zscheischler et al. (2020) |
| Triggering/ cascades | "One hazard causes another hazard to occur, which can result in hazard chains, networks, or cascades." | Ciurean et al. (2018) |
| Amplifying | "The occurrence of one hazard can increase the likelihood and/or magnitude of additional hazards in the future." | Ciurean et al. (2018) |
| Indicator | "Observable and measurable characteristics that can be used to simplify information to help understand the state of a concept or phenomenon, and/or to monitor it over time to show changes or progress towards achieving a specific change." | Gill et al. (2022) adapted from Ivčević et al. (2019) |
| Vulnerability | "The conditions determined by physical, social, economic and environmental factors or processes which increase the susceptibility of an individual, a community, assets or systems to the impacts of hazards." | Sendai Framework Terminology on Disaster Risk Reduction (UNDRR, 2015) |
| Impact | The realised, or potential consequences on natural and human systems, where consequences result from the interactions of hazards, exposure, and vulnerability. Impacts generally refer to effects on lives, livelihoods, health and well-being, ecosystems and species, economic, social and cultural assets, services (including ecosystem services), and infrastructure. Impacts may be referred to as consequences or outcomes. | IPCC (2018) |
| Qualitative method approach | "Qualitative research methods aim to address societies' scien-tific and practical issues and involve naturalistic and in-terpretative approaches to different subject matters. These methods utilize various empirical materials such as case studies, life experiences, and stories that show the routines and problems that individuals are struggling with in their | Taherdoost (2022) |



| | lives through focusing on their in-depth meaning and mo-tivations which cannot be defined by numbers. Qualitative research aims to collect primary, first-hand, textual data and analyse it using specific interpretive methods." | |
|---|---|---|
| Quantitative method approach | "Quantitative research methods aim to define a particular phenomenon by collecting numerical data to address specific questions such as how many and what percentage in different fields. It is the method of employing nu-merical values derived from observations to explain and describe the phenomena that the observations can reflect on them. This method employs both empirical statements, as descriptive statements about the meaning of the cases in real words not about the ought of the cases, and methods. It also applies the empirical evaluations intending to determine to which degree a norm or standard is fulfilled in a particular policy or program. Finally, the collected numerical data is analysed using mathematical methods." | Taherdoost (2022) |
| Mixed-method approach | "Mixed-method methods simply employ a combination of both qualitative and quantitative approaches based on the purpose of the study and the nature of the research question aiming to provide a better understanding of the subject." | Taherdoost (2022) |


To date, there has been no concerted effort to collate and review existing multi-hazard and multi-risk indicators
or attempt to unify these approaches and demonstrate their potential value in DRR activities. This paper uses a
systematic review process to document and explore the use of indicators within the multi-hazard and multi-risk
contexts for the first time and sets out recommendations for their future development and use. The review paper
is structured as follows: section 2 lays out the methodology for the systematic literature review and the analysis
of the findings; section 33 provides a detailed overview of the use of indicators in hazard and risk assessments;
section 4 provides a wider discussion and a suggested recommendations for the expansion and use of multi-hazard
and multi-risk indicators; and section 5 provides some concluding remarks.
**2.  Methods**
A systematic literature review approach was employed to identify peer-reviewed literature that either use
indicators, or analyse the use of indicators, in multi-hazard and multi-risk studies, guided by the Preferred
Reporting Items for Systematic reviews and Meta-Analyses (PRISMA) protocol (Page et al., 2021). The
methodology followed six steps: 1) definition of key search terms, 2) identification of records, 3) screening of
results based on inclusion and exclusion criteria, 4) categorising the research papers into two broad categories of
multi-hazard and multi-risk studies, 5) selecting key works from each category that are the most significant and



provide good examples of multi-hazard and multi-risk indicator use, 6) assessing the suitability of each record in
more detail.
The Scopus, Web of Science and PubMed databases were used to extract literature related to indicators in multi-
hazard and multi-risk studies, due to their comprehensive coverage of peer-reviewed articles. The search terms
(Table S1) were stratified into two levels. The first level encompassed terminology associated with multi-hazard
and multi-risk studies, including alternative spellings and descriptors such as "compound", "interacting",
"cascading", and "interconnected" hazards and/or risks. A total of 22 Level 1 search terms were employed. The
alternative terminologies were combined using an "OR" Boolean operator and then paired with Level 2 search
terms using an "AND" Boolean operator. Level 2 comprised five search terms related to indicators and alternative
or related terminology for indicators (i.e., "index", "indices", "metric", "disaster risk indicator"). The search terms
were applied across title, abstract, and keywords. Although not exhaustive, this set of search terms effectively
narrowed the research scope to multi-hazard and multi-risk studies, excluding single hazard or risk papers that fall
outside the scope of this study. The search strings used across all three databases, together with relevant keywords
and Boolean operators, are provided in Table S2.
The initial search returned 1468 articles that met the search criteria. A date restriction was applied to include only
papers published post-2015, aligning with the publication year of the Sendai Framework for Disaster Risk
Reduction and its emphasis for the adoption of a multi-hazard approach. A duplicate removal process, executed
using R, was applied to the 1,140 articles, identifying and excluding 515 duplicates. Fig. 2 provides a flowchart
detailing the screening process, including the number of articles at each stage of the review.

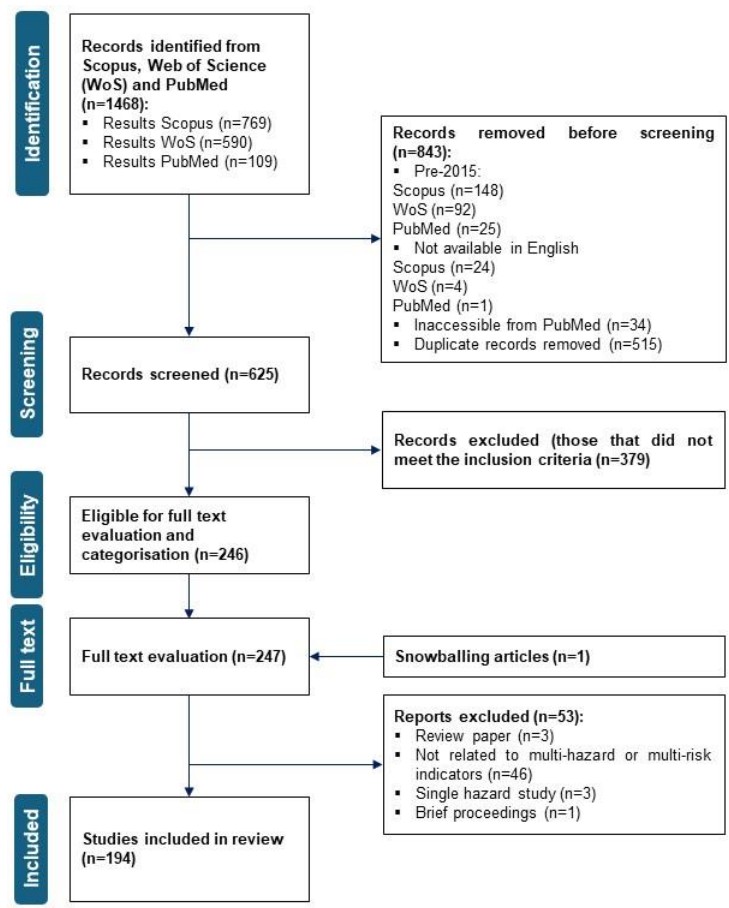

**165**
**166**

**167** **Figure 2. Flowchart of the systematic literature review used in this study, showing the identification, screening and**
**168** **inclusion process together with the numbers of articles at each stage.**

**169**

**170** After removing duplicates, a two-part screening process was applied to the remaining unique 625 articles. Initially,
**171** all articles were screened based on their titles and abstracts to create a database comprising papers considered
**172** relevant for further review, while irrelevant papers were excluded. Relevance was primarily assessed based on the
**173** use of multi-hazard and multi-risk indicators in evaluating natural hazards of geo and hydrometeorological origin
**174** across diverse research domains. The first phase of the screening excluded 379 papers, leaving 246 that were
**175** relevant for further investigation. In the second screening phase, the full text of these 246 articles were evaluated.
**176** An additional reference (i.e., snowballing article) was identified and included during the full text evaluation
**177** (n=247) stage. A database was established to collect the retrieved information (Pickering and Byrne, 2014) and
**178** to minimize the risk of bias in the selection process. A total of 53 articles were excluded at the full-text evaluation
**179** stage. The following exclusion criteria were applied during both screening phases:

**180** • Articles that did not align with the study's objectives, as determined by the title, abstract, or
**181** keywords.




• Review articles.
• Studies focusing on risks related to animal, bird, plant species, marine habitats, human health,
pollution, unmanned vehicles, workplace safety, finance and insurance, and nuclear risks.
• Studies investigating structures, electrical grids, infrastructure resilience, and transport networks
in terms of robustness, functionality, or performance based on structural integrity or design.
• Articles that did not address or utilise multi-hazard or multi-risk indicators.
• Brief conference proceedings.
Following the screening process (i.e., full text evaluation), the remaining 194 papers were analysed and critically
assessed. These papers were used to extract information on single hazard types, categories of single hazards
according to UNDRR hazard information profile (Murray et al., 2021), as well as on vulnerability, impact and
risk assessment approaches, including quantitative, qualitative and mixed methods. The terms "risk" and "impact"
were both employed in this analysis to encompass studies that focus on potential future consequences, typical of
risk assessments, as well as those associated with past events. Additionally, the exposure category was not
evaluated separately, as it is implicitly considered in the adopted vulnerability typologies and the consequences
evaluated in the risk/impact assessments. Definitions of the different types of approaches are provided in Table 1.
Two categories of studies were identified:
**Category 1**: Multi-layer single hazard and risk – these papers individually analysed multiple single hazards
or risks occurring in a certain location and overlay the outcomes. Although these types of assessments were
termed multi-hazard or multi-risk by the authors, the hazards were analysed individually and therefore not
considered multi-hazards as per the Category 2 definition in this paper.
**Category 2**: multi-hazard and multi-risk – herein interactions between hazards were considered. These
studies were further categorised into two broad classes based on the type of interaction: compound; and
triggering and amplification studies. Definitions of different types of multi-hazard interactions or
interrelationships are listed in Table 1.
The multi-hazard and multi-risk studies identified were further reviewed to extract information on the indicators.
Indicators were categorised into five classes according to their use to describe hazard characteristics such as
intensity, frequency, and probability (UNDRR, 2017a), and to develop composite indicators, and finally, studies
with no indicator.
**3.  Results**
**3.1 Findings from the articles reviewed**
**3.1.1 Distribution of articles with respect to various risk–related components**
The majority of the articles (88%) focused on hazard assessment, followed by vulnerability and risk/impact
assessments, as demonstrated in Fig. 3. In terms of vulnerability and impact, 49% of the articles conducted
vulnerability assessments, while 35% included risk/impact assessments. We also analysed the different
combinations of risk-related components (Table 2). Approximately 40% of the articles explored both hazard and
vulnerability, indicating a significant overlap between these two areas. In contrast, only 6% of the articles
considered both hazard and risk/impact, suggesting that direct linkages between these components were less





frequently examined. However, 29% of the articles addressed both vulnerability and risk/impact, highlighting the
importance of understanding how vulnerabilities translate into tangible risk/impacts. Additionally, 20% of the
articles integrated all three components – hazard, vulnerability, and risk/impact – demonstrating the complexity
and interdisciplinary nature of a significant portion of the research.

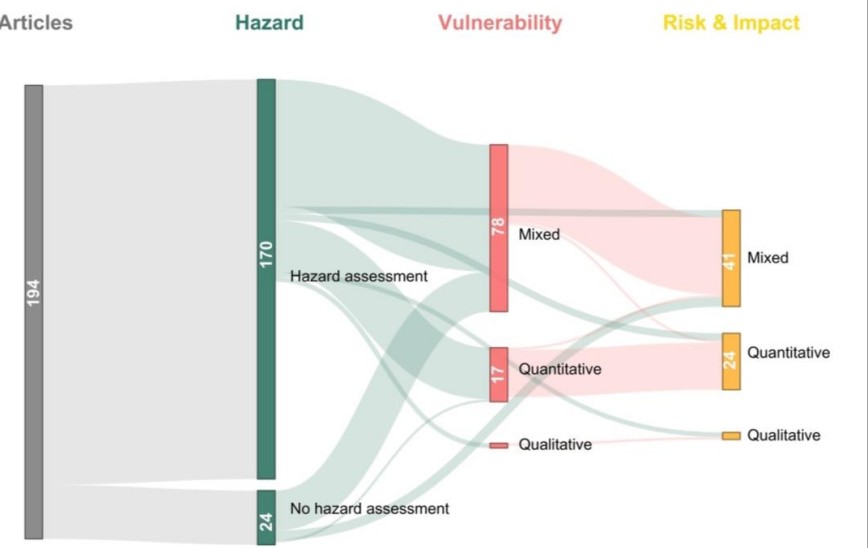

**Figure 3. Sankey diagram illustrating the evolving focus in multi-hazard studies (grey), highlighting a gradual shift**
**from exclusive hazard analysis (green) to the integration of vulnerability components (red), and finally, to the inclusion**
**of risk and impact (orange). The white numbers indicate the number of articles associated with each node in the**
**diagram, and the flow dimensions are proportional to the number of articles transitioning between nodes.**

In terms of the methodologies used (Table 3), we found that for hazard assessment 88% of the articles employed
quantitative methods, indicating a strong preference for numerical and statistical approaches in this area. The
methodologies used for vulnerability assessment were more varied: 24% of the articles used quantitative methods,
2% adopted qualitative approaches, and the majority (74%) employed a mixed-methods approach, integrating
various analytical techniques. For risk/impact assessment, 35% of the articles applied quantitative methods, 4%
used qualitative methods, and 60% employed a mixed-methods approach. This suggests that the integration of
multiple methodologies was considered essential for a comprehensive understanding of the potential/realized
consequences of risk/impact.
**Table 2. Matrix showing the percentage of papers that consider single or multiple risk components.**

|  | **Hazard** | **Vulnerability** | **Impact** | Combined |
|---|---|---|---|---|
| Hazard | 88% | 40% | 6% |  |
| Vulnerability |  | 49% | 29% | 20% |
| Impact |  |  | 35% |  |




**Table 3. Examples of single-hazard indicators used in risk analyses.**

|  | No | Quantitative | Qualitative | Mixed |
|---|---|---|---|---|
| Hazard | 12% | 88% | 0% | 0 |
| Vulnerability | - | 24% | 2% | 74% |
| Impact | - | 35% | 4% | 60% |

### 3.1.2 Distribution of articles according to hazard interactions

The 194 articles reviewed in this study analysed a total of 493 individual hazards. We analysed the frequency of different hazards and their interactions (Fig. 4), finding that meteorological and hydrological hazards are the most frequently studied, accounting for 63% (311) of the total hazards, followed by geohazards (22%), environmental hazards (10%), and technological hazards (2%). Notably, in 15 instances (3%), no individual hazard was considered. We also found that although all articles focused on multi-hazard events, the majority do not analyse interactions between hazards. These hazards were categorised as multi-layer single hazards, accounting for 53% (260) of the total hazards analysed. In these cases, multiple single hazards were analysed individually, without considering their interactions in time and/or space. Compound interactions were the second most common, representing 29% (145) of the total hazards, where multiple hazards occurred simultaneously or in close sequence. Triggering and amplification interactions accounted for 12% (57) of the hazards, where one hazard might trigger or amplify the effects of another. Finally, in 6% (31) of the cases, no interaction between hazards was identified.

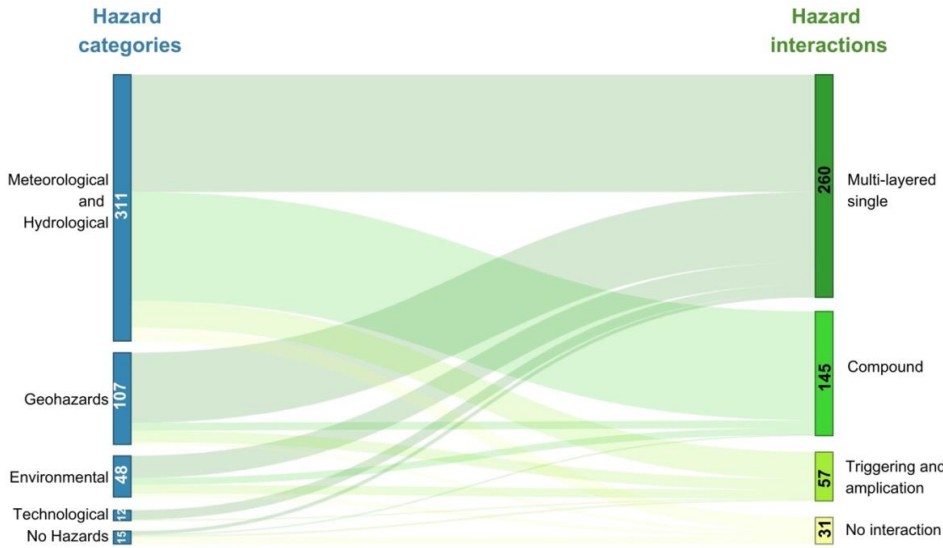

**Figure 4. Sankey diagram illustrating the distribution categories (blue) and interactions (green) of 493 hazards analysed across 194 research papers. The numbers indicate the number of hazards associated with each node in the diagram, and the flow dimensions are proportional to the number of hazards transitioning between nodes.**



Nearly all compound interactions originated from meteorological and hydrological hazards (88%), underscoring
the tendency of these events to co-occur with other hazards in a combined manner (Fig. 4). In contrast, geohazards
exhibited a different pattern. The majority of geohazards were associated with multi-layer single hazard
interactions (77%), indicating that these hazards were often studied as recurring or overlapping events rather than
as part of complex interactions with other hazards. When it came to multi-hazard interactions, geohazards were
almost equally distributed between compound (8%) and triggering and amplification interactions (12%). Finally,
technological hazards displayed a distinct trend where all instances were categorised under layered single hazard
interactions, suggesting that these hazards were primarily analysed as isolated incidents, without significant
consideration of their potential to compound with or trigger other hazards.
**3.2 Multi-layered single hazard and risk indicators**
**3.2.1 Multi-layered single hazard indicators**
We found that approximately 44% of the 194 articles reviewed were categorised as multi-layer single hazard
studies, predominantly focusing on meteorological, hydrological, and geo hazards (Fig. 4). In some instances,
weights were assigned to individual hazard layers to reflect their relative importance. However, interactions
between hazards were not considered in any of these cases. Indicators used in multi-layer single hazard studies
were either applied to individual hazards or combined into composite indicators (index). Indicators that describe
hazards on an individual basis tend to define the hazard in terms of its extent and intensity. Single-variable
indicators, which help define the occurrence of a hazard, are commonly employed in Machine Learning (ML)
applications to identify potential locations where a particular hazard may occur. This is prevalent in studies
assessing flood and landslide susceptibility, where past occurrences and specific hazard pre-conditioning factors
(indicators) are used to determine other areas with future hazard potential or susceptibility (Nguyen et al., 2023;
Rehman and Azhoni, 2023; Pourghasemi et al., 2020). These types of studies rely heavily on the quality of the
input data and often face challenges related to insufficient evidence regarding the interactions among diverse
hazards. Although they serve a purpose in identifying areas susceptible to hazards, they frequently overlook the
interactions between different hazards and their temporal overlaps.
In multi-layer single hazard studies where composite indicators were used, multiple hazard maps for a specific
geographical region were often overlaid. For example, Emrich et al. (2022) presented a Composite Multi-Hazard
Index (CHI) map, combining 15 natural hazards in the USA and classifying them into five hazard groups: (1)
severe weather, (2) flooding, hurricanes, and storm surges, (3) winter weather, (4) heat, drought, and wildfires,
and (5) earthquakes. Such composite indicators were generally standardized or normalized to categorise them into
various scales, such as 'very low' or 'very high' hazard intensities. This approach allows for the inclusion of a large
number of variables in the analysis, and there are many examples of its use in the development of hazard maps
(e.g., Wang and Sebastian, 2022; Fleming et al., 2023; Ou et al., 2022; Barasa et al., 2022; Murnane et al., 2019).
However, since this approach does not account for hazard interactions, overlaying of multiple hazard maps is not
considered a comprehensive multi-hazard approach.
The multi-layer single hazard approach also includes studies that assign weights to individual hazard causative
factors, showing variations in the significance of hazard intensities and developing multi-hazard indicators (Liu et
al., 2016). For instance, Analytical Hierarchy Process (AHP) (Durlević et al., 2021; Guerriero et al., 2022) and



Machine Learning-based algorithms (Mandal et al., 2022) were widely used to estimate such weights and develop multi-hazard indicators. Nevertheless, these approaches also ignore interactions between hazards in space and time. Therefore, this study categorised them as multi-layer single hazard studies.

**3.2.2 Risk indicators based on multi-layer single hazards**

In recent years, risk assessment and management research has increasingly focused on the analysis of multi-layer single hazards. Among the 86 studies analysing multi-layer single hazards, we found that approximately 41% (n=35) addressed risk, with the majority of these (n=32) focusing on meteorological and hydrological hazards.

Risk studies related to multi-layer single hazards were conducted at various scales, from global to national levels, each offering methodologies suited to their specific contexts. For example, Marulanda Fraume et al. (2020) presented a holistic assessment framework at the global level using data from 216 countries. This framework evaluated physical risks based on potential damages directly linked to individual hazard occurrences, while also assessing underlying risk drivers and amplifiers. At the national level, Zuzak et al. (2022) introduced the National Risk Index (FEMA US), a multi-hazard risk measurement approach that used diverse geographic data sets and risk factors to provide a national overview of risk at the county level. This index relied on a robust transdisciplinary methodology, incorporating direct stakeholder involvement and various composite indicators to produce an integrated risk assessment.

There are notable methodologies for multi-hazard risk assessment that integrate exposure to various natural hazards along with considerations of social vulnerability. For instance, Bixler et al. (2021) developed a quantitative social vulnerability indicator adapted from the Social Vulnerability Index (SoVI), combined with hazard exposure. This assessment was conducted by analysing individual hazards spatially at the census block scale. Similarly, Guillard-Gonçalves et al. (2015) estimated a SoVI to delineate vulnerable and risk zones for six natural hazards in Greater Lisbon, including earthquakes, floods, flash floods, landslides, tsunamis, and coastal erosion. The study used susceptibility maps and population exposure data to develop a risk matrix and map at the parish level.

Existing studies on multi-layer single hazard risk assessment have also considered various assets, ranging from cultural heritage sites to rural communities, agricultural sectors, and socio-economic and infrastructure systems. For example, Valagussa et al. (2021) investigated the risks posed by multiple hazards to cultural heritage sites in Europe, introducing the UNESCO Risk Index, which integrates hazard and potential damage considerations to prioritize interventions. Asare-Kyei et al. (2017) quantified the risk and vulnerability of rural communities in West Africa to drought and floods, developing the West Sudanian Community Risk Index, which was validated through a novel Community Impact Score (CIS).

The multi-criteria decision-making approach (MCDA) is widely used to estimate risk from multiple natural hazards. For instance, Arvin et al. (2023) quantified exposure to floods, earthquakes, and landslides, alongside infrastructure resilience, at a regional scale in Iran, integrating 25 quantitative indicators using the MCDA. Similarly, Pagliacci (2019) introduced a risk assessment framework for the Italian agri-food sector, using composite indicators to represent hazards, exposure, vulnerability, and risk across various municipalities. Nofal et al. (2023) proposed a methodology for assessing the resilience of buildings, power, and transport infrastructure to hurricane-induced hazards, incorporating factors such as physical damage, functionality, and demographic changes. This study also used a composite indicator to explain the extent of structural damage caused by multiple hazards. Additionally, Asare-Kyei et al. (2017) developed a community-based socio-ecological-systems (SES)





indicator to assess risk and vulnerability at the community level, introducing a CIS for validation. Anderson et al.
(2021) assessed vulnerability to hazards in the Mississippi Delta, calculating separate ecological and social
vulnerability indices, which were then integrated to create a multi-hazard vulnerability index.
**3.3 Multi-hazard and multi-risk indicators**
**3.3.1 Compound hazard indicators**
Among the 493 different types of hazards included in the 194 reviewed articles, we identified about 29% (n=145)
as compound multi-hazard events (Fig. 4). Various indicators were used to explain these compound events.
Composite indicators were the most common, used to explain 50% of all hazards, followed by probability (16%),
frequency (16%), and intensity (3%). Notably, 14% of the total compound multi-hazard events were not associated
with any specific hazard indicators (Fig. 5a). These studies primarily employed different types of multi-risk
indicators.

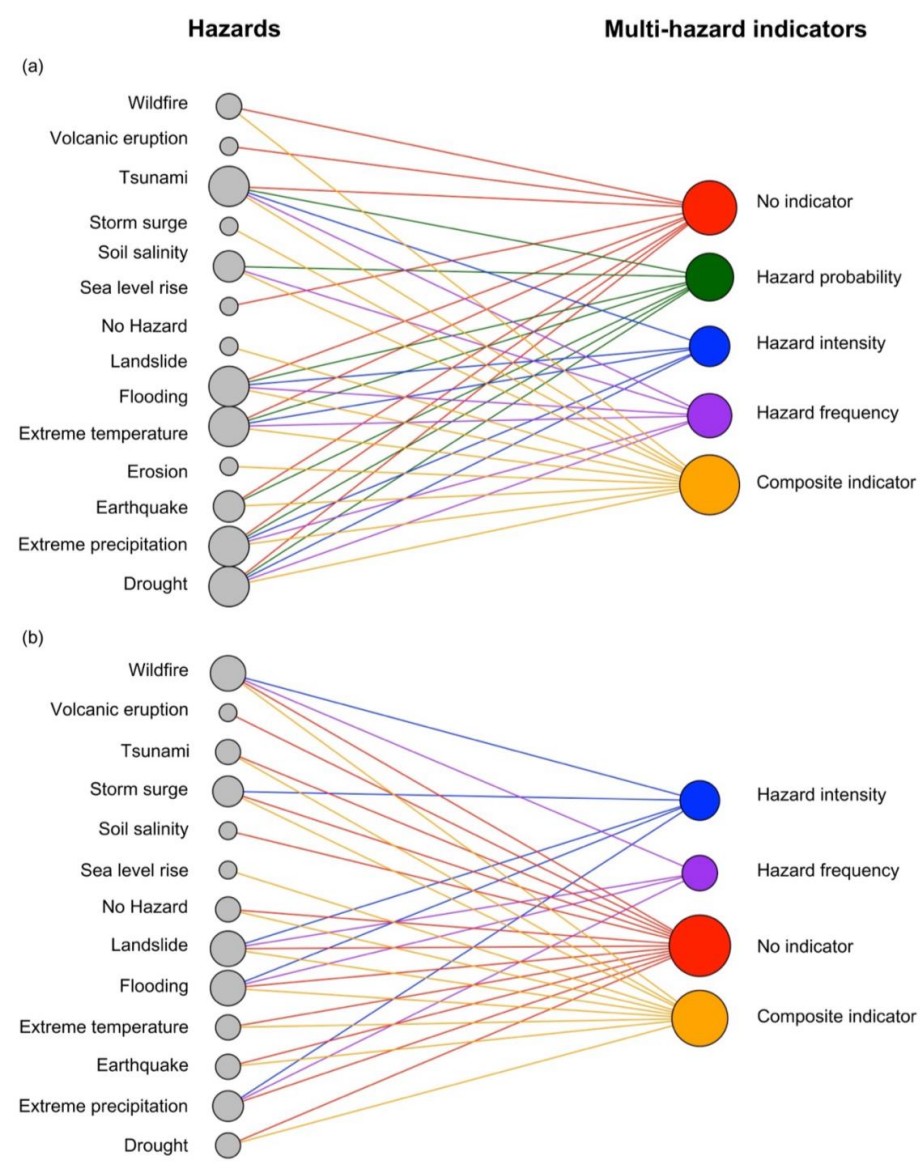

**Figure 5. Bipartite graph illustrating the relationships between 14 natural hazards and five multi-hazard indicators for (a) compound multi-hazard and (b) triggering and amplification events. The connections represent the extent to which specific indicators are applied in the assessment of different hazards.**

We observed a diverse range of compound multi-hazard indicators across the studies. Studies focused on meteorological hazards, for example, such as drought and extreme temperatures, primarily used composite indicators. However, the development and application of these indicators varied. Feng et al. (2021) evaluated compound dry and hot events (CDHEs) across different global regions in maize-producing areas. Their study



utilized three different single-hazard drought indices—the self-calibrating Palmer Drought Severity Index
(scPDSI), the Standardized Precipitation Index (SPI), and the Standardized Precipitation-Evapotranspiration Index
(SPEI)—along with the Standardized Temperature Index (STI) as "hot" indicators. Bonekamp et al. (2021)
combined individual and multi-hazard indicators to assess extreme temperature and precipitation under present-
day and future climate change scenarios using a total of seven indicators, five of which were single-hazard
indicators, while the remaining two were multi-hazard indicators associated with spatially and temporally
compounding events.
Some studies developed specific compound multi-hazard indicators. For instance,  Bian et al. (2022) developed
the Compound Drought Heatwave Magnitude Index (CDHMI) to investigate compound drought-heatwave (CDH)
events in eastern China. This index was based on the non-stationary Standardized Precipitation Evapotranspiration
Index (NSPEI) and daily maximum temperature (Tmax) to determine the probability of heatwave and drought
events exceeding normal thresholds, reflecting the intensity of such composite events to some extent. Qian et al.
(2023) also applied a CDHMI but used a heatwave magnitude index and a drought magnitude index instead of
Tmax and SPEI data. Another indicator for hot and dry climatic events, the Dry-Hot Magnitude Index (DHMI),
was developed by Wu et al. (2019) to characterize the magnitude of compound dry and hot events using monthly
precipitation and daily maximum temperature data.  Hao et al. (2019) proposed the Standardized Compound Event
Indicator (SCEI), which integrates both dry and hot conditions, representing their severity. This study
characterized drought and hot conditions using the two standardized indicators SPI and STI. This indicator was
also used alongside the El Niño–Southern Oscillation (ENSO) in a model to predict compound hot-dry events.
In studies related to hydrological hazards, such as storms and floods, composite or probabilistic indicators were
primarily used. For instance, to investigate terrestrial and coastal flooding events, Jalili Pirani and Najafi (2022)
employed a statistical approach (copula theory) to derive dependencies between multiple drivers of flooding, such
as extreme precipitation, river overflows, and storm tides. Additionally, they used a Compound Hazard Ratio
(CHR) index to characterize the interactions between different drivers and their effects on return level estimates
of compound events. Ganguli and Merz (2019) analysed spatiotemporal trends in compound flood events caused
by the co-occurrence of fluvial floods and extreme coastal water levels to understand historical trends in compound
flooding in NW Europe from 1901 to 2014, using a CHR Index that links fluvial discharge with coastal water
levels. Mitu et al. (2023) developed a new topographic indicator (D-Index) to identify surge-dominated, flow-
dominated, and compound-dominated areas. Alberico and Petrosino (2015) recognised the need to develop two
indices for multi-hazard events based on their recurrence intervals, demonstrating the link between time and
hazard. These indices were constructed based on the time window considered for hazard assessment or the
probability of the hazard occurring.

### 3.3.2 Triggering and amplification hazard indicators

Of the approximately 12% (n=57) of the 493 hazards (multi-layer single hazards and multi-hazard events)
identified as triggering and amplification types of multi-hazard events (Fig. 4), nearly half of these (49%) were
not associated with any specific hazard indicators. These studies also utilised various indicators to explain
vulnerability and impacts. Composite indicators were the most commonly used, explaining 38% triggering and
amplification events, followed by frequency (7%) and intensity (6%). Notably, probability indicators were not
used for this type of hazard (Fig. 5b).




Triggering and amplification events were prominent across various meteorological, hydrological, and geo hazards.
Different approaches were employed to develop composite indicators for meteorological hazards. For example, in
the context of fire events, Khorshidi et al. (2020) considered fire size as an indicator to explain the co-occurrence
of wildfires that triggered 'megafires,' rather than extreme magnitudes of individual drivers or their additive
combinations. They investigated correlations between eight individual drivers and various fire sizes, analysing
data from southern California counties. Similarly, Piao et al. (2022) aimed to create a map identifying areas highly
prone to forest fires where forest fires and droughts spatially coincide; by leveraging existing quantitative
indicators like the Standardized Precipitation Index (SPI) and the Enhanced Vegetation Index (EVI) alongside
machine learning techniques, they produced composite indicators.
In studies involving geohazards, Bernal et al. (2017) conducted a risk assessment for multi-hazard occurrences in
Manizales, Colombia. They considered probabilistic hazard maps produced for single hazards, such as
earthquakes, volcanic (lahar), and landslide hazards. Landslide susceptibility was determined through an artificial
neural network, with earthquakes and extreme rainfall as triggering factors, each with its own threshold. A risk
assessment was conducted, involving a quantitative and probabilistic relationship assessment between different
hazard intensities and a mean damage ratio. In the context of hydrological hazards, Rocha et al. (2021) developed
a flood risk management framework using hazard data and risk indicators to determine coastal vulnerability for
the western Portuguese coastal zone. This study considered coastal flooding as a precursor to coastal erosion.
Regarding the combination of geo and hydrological hazards, Coscarelli et al. (2021) investigated the relationship
between climate indices and the frequency of landslides, floods, and forest fires in Italy. Using climate indices
derived from local weather stations, they compared past hazard occurrences with climate indices to develop
predictive models. Their findings suggest that landslides were more associated with moderate rainfall, floods with
extreme rainfall, and forest fires with moisture content in the fuel. Argyroudis et al. (2019) studied consecutive
earthquake and flood hazards to develop a multi-hazard resilience index, considering "damage state to bridge" as
an indicator. Ramli et al. (2021) developed an Integrated Disaster Risk Assessment Framework (IDRAF), building
on and expanding the theoretical framework established by the EU Project MOVE (Multi-Hazard Spatial
Overlapping and Methods for the Improvement of Vulnerability Assessment in Europe). The IDRAF encompassed
eight meteorological, hydrological, and geo hazards, with the characterization of multi-hazard scenarios involving
two key components: frequency of occurrence and spatial interaction. Risk quantification was facilitated through
a multi-hazard, multi-vulnerability approach, with indicators determined via a semi-quantitative methodology. The
framework's applicability was demonstrated through its implementation at a local administrative scale in Malaysia,
with evaluation conducted by 64 local experts in disaster risk management representing various governmental and
non-governmental entities.
**3.3.3 Multi-risk indicators**
Among the 90 studies related to multi-hazards – including compound, triggering, and amplification hazards – we
found that 31% (n=28) analysed risks or impacts. The overall risk calculation varied across studies, with different
combinations of risk metrics such as hazard, exposure, vulnerability, coping capacity, susceptibility, sensitivity,
and resilience commonly used. Table 4 summarises the various indicators used to explain risks and their
components.
**Table 4. Examples of various indicators used in risk analyses.**



|  | Indicator | References |
|---|---|---|
| Hazard | • Coastal flooding: time horizon, probabilistic sea level rise height, projected emissions levels, risk a version, storm frequency<br><br>• Stormwater flooding: flow accumulation, rainfall intensity, geology, land use, slop, elevation, distance from drainage network, percent of land area below highest observed storm surge water level<br><br>• Landslides: Percent of land area with steep slope (higher than 45 degrees)<br><br>• Erosion: water erosion, wind erosion<br><br>• Drought: drought severity, drought coverage<br><br>• Heatwaves: annual days over 90°F, occurrence of very hot days (a maximum temperature greater than or equal to 33°C) and hot nights (a minimum temperature greater than or equal to 28°C)<br><br>• Wildfire: fire frequency, potential fire behaviour | Johnson et al. (2016); Fleming et al. (2023); Ghosh and Mistri (2022) |
| Exposure | • Socioeconomic data (Populations, education)<br><br>• Topographical factors<br><br>• State of buildings (e.g., construction and occupancy)<br><br>• Land use (e.g., forest, buildings, agriculture)<br><br>• Infrastructure (e.g., roads, railways, bridges, powerlines) | Johnson et al. (2016); Fleming et al. (2023); Ghosh and Mistri (2022); Haque et al. (2020); Jacome Polit et al. (2019); Sahana and Sajjad (2019); Viavattene et al. (2018); Depietri et al. (2018); Sekhri et al. (2020); Barasa et al. (2022); Murnane et al. (2019) |
| Vulnerability | • Exposure, susceptibility and lack of coping capacity<br><br>• Exposure, sensitivity and adaptive capacity<br><br>• Exposure sensitivity and resilience<br><br>• Socioeconomic (e.g., poverty, employment, access to communications, transport and services)<br><br>• Physical vulnerability (people or assets)<br><br>• Social vulnerability (education and food security)<br><br>• Natural resources<br><br>• Risk to life |  |





| Coping capacity/lack thereof | • One person households<br>• Language barrier<br>• Governance<br>• Living conditions (e.g., access to mobile phone, electricity, sanitation, water)<br>• Health (e.g., healthcare expenditure, maternal mortality rate, diet, access to healthcare) | |
|---|---|---|
| Adaptive capacity | • Education (e.g., literature rate)<br>• Economic status (e.g., employment)<br>• Health (e.g., access to medical services)<br>• Living conditions (access to banking, electricity, mobile phones) | |
| Sensitivity | • Population descriptors (e.g., density, age, sex)<br>• Living conditions (e.g., housing type)<br>• Economic conditions (e.g., employment)<br>• Land coverage (e.g., built-up areas, agricultural Building material and condition) | |
| Susceptibility | • Population descriptors<br>• Economic status (e.g., employment, income) | |
| Resilience | • Living conditions (e.g., housing type, access to banking services, access to mobile connection) | |


We noted that existing studies generally defined risk by overlaying various indicators of vulnerability, exposure,
and coping capacity to create vulnerability or risk indices (Beltramino et al., 2022). For example, the Cumulative
Vulnerability Index is an approach that functions as a composite of exposure, sensitivity, and adaptive capacity
indicators (Krishnan et al., 2019). Similarly, an Integrated Coastal Vulnerability Index was developed by
combining multiple sub-index parameters, including coastal characteristics or physical variables, wave or coastal
forcing, and socio-economic factors (Godwyn-Paulson et al., 2022; Ariffin et al., 2023; Hoque et al., 2019).
Another commonly used vulnerability indicator is the Social Vulnerability Index (SoVI), which combines various
socioeconomic and built environment variables to quantify social vulnerability (Cutter et al., 2003). Although
some studies applied the SoVI approach to multi-hazard events, its methodology is hazard-agnostic, allowing for
interchangeable hazards. For instance, Yang et al. (2015) developed a SoVI to quantify regional social
vulnerability to natural hazards and mapped its spatiotemporal distribution in China. Socioeconomic variables
were used as indicators; however, no specific natural hazard or impact was defined. This hazard-agnostic approach
has also been applied in other risk assessments that did not use the SoVI approach. These composite indicators
helped create single indicators, simplifying numerous data inputs and providing an efficient method for assessing





certain parameters. They were particularly useful in complex systems where one indicator was insufficient to
explain multiple variables (Marulanda-Fraume et al., 2022).
More recently, multi-risk indicators such as Social-Ecological Systems (SES) have been developed, incorporating
variables related to hazard, exposure, and vulnerability. For instance, Ou et al. (2022) applied the SES framework
in deltaic regions to assess multi-hazard risks (e.g., cyclones, floods, storm surges, and droughts) by calculating a
Global Delta Risk Index (GDRI). However, Ou et al. (2022) did not consider interactions between hazards when
estimating risks. Similarly, Cremin et al. (2023) used the GDRI to assess socio-ecological systems in river deltas,
allowing for a better understanding of ecosystem exposure, sensitivity, and robustness.
Several studies have also developed libraries of multi-risk indicators. These databases allow users and stakeholders
to review, select, and customize indicators for specific needs. Such libraries typically cover indicators related to
social, ecological, and economic factors for various hazards and local contexts (Shah et al., 2020; Sebesvari et al.,
2016). For instance, Hagenlocher et al. (2018) created a library of hazard-dependent and hazard-independent
vulnerability indicators, providing users with access to indicators that can be applied in specific contexts and for
specific hazard types relevant to deltas.
The integration of multiple layers of risk indicators can introduce uncertainties, particularly when equal weights
are assigned to each risk component. As a result, we found several recent studies that have developed methods to
estimate weights for each indicator, reflecting the varying significance of different risk parameters. Expert
judgment is a commonly applied method for estimating these weights (Mafi-Gholami et al., 2019; Arvin et al.,
2023; Cotti et al., 2022). For example, Gallina et al. (2016) used multi-hazard weighted scores generated through
influence weightings in a hazard matrix to evaluate multi-risk. However, expert judgment-based weight
calculations can also be subject to systematic bias (Jacome Polit et al., 2019).
**4.  Discussion and recommendations for development and implementation**
**4.1 Key findings**
In a context where United Nations' member states are increasingly advocating for multi-hazard approaches, multi-
hazard risk data, and multi-hazard risk governance (United Nations, 2023), there is likely to be a growing demand
for aligned indicators and indicator-informed approaches that support both implementation and monitoring of the
Sendai Framework. The increase in research activity demonstrated in this review is in-part explained by this policy
demand, with a succession of European Union-funded research projects focused on multi-hazards and multi-risks.
These and other ongoing projects have been established to investigate the challenges posed by multi-hazards and
multi-risks, highlighting a clear momentum towards a shift from single to multi-hazard analysis and multi-risk
assessment and management.
Our review has highlighted the broad use of indicators for risk assessment and management (i.e., Bernal et al.,
2017; Sekhri et al., 2020), to identify interactions between hazards (i.e., Jalili Pirani and Najafi, 2022), and as
stand-alone indicators for establishing warning thresholds (i.e., Vitolo et al., 2019; Li et al., 2021). However, this
study finds that there are few studies that explicitly develop indicators for multi-hazard and multi-risk contexts.
Through our review and analysis of these indicators, we note the following:



- The selection and use of different terminology and definitions by different groups affects the development and use of indicators and remains a challenge for the advancement of multi-hazard risk work (Sections 1 and 2).

- While there are many useful examples of indicators being developed and used in layered single hazard studies, the global hazard and risk literature also recognises that interrelationships exist between hazards, and between hazards and other risk components. These interrelationships should be considered in indicators to advance multi-hazard risk work (Section 3.1.2).

- Current work on indicators supporting multi-hazard risk management is dominated by a focus on compound event type interrelationships, with less work on indicators for triggering and amplification type interrelationships. Indicators for triggering and amplification type interrelationships require understanding of the physical processes coupling two or more hazards (Sections 3.3.1 and 3.3.2).

- Research on hazard assessment was found to be more common than studies on other components of risk (e.g., vulnerability) or broader characterisation of risk itself. There are limited examples of multi-risk indicators that embed understanding of multi-hazard interrelationships (Sections 3.1.1 and 3.3.3).

- The findings reveal a lack of stakeholder engagement and prioritisation in developing multi-hazard multi-risk indicators; the extent to which these can therefore translate effectively into supporting multi-hazard disaster risk management is ambiguous (Section 4.1).

Aspects of these findings align with similar studies on the increase in multi-hazard literature. For example, with respect to the impact of terminology and varying interpretations of multi-hazard concepts, Kappes et al. (2012) noted the diversity of terms used for hazard interrelationships, Gill and Malamud (2014) reflect on the impacts of different interpretations of the multi-hazard concept (the multi-layer single hazard perspective vs. a more holistic multi-hazard approach), and Ciurean et al. (2018) reviewed different classifications of hazard interrelationships before synthesising these into a proposed taxonomy (subsequently adopted in Gill et al. (2022)). The impact of variations in terminology is evident in the development and application of indicators. Risk management would be strengthened by the creation of and adherence to guidance for the development and use of indicators in multi-hazard, multi-risk contexts, building on existing good practices and drawing on established and agreed terminology and definitions.

The broader multi-hazard literature also demonstrates a wide array of new and developing methods for characterising hazard interrelationships (e.g., Gill and Malamud, 2014; Tilloy et al., 2019; Zscheischler et al., 2020; De Angeli et al., 2022; Claassen et al., 2023; Lee et al., 2024) and dynamics of other components of risk (e.g., De Ruiter and Van Loon, 2022; Hochrainer-Stigler et al., 2023). A breadth of approaches is likely necessary to support risk characterisation in different contexts (e.g., data poor vs. data rich), but variation in the approaches used to characterise hazard interrelationships may make it challenging to develop generic indicators for monitoring the management of multiple, interrelating hazards and their associated risk.


## 4.2 Developing effective multi-hazard and multi-risk indicators: challenges and opportunities

The results from this systematic review show the use of indicators in multi-layer single hazard and multi-hazard and multi-risk contexts. Indicators are selected and used to help characterise interactions between hazards (including the probability of multi-hazard interrelationships, the co-occurrence of multi-layer single hazard events, and changes of multi-hazard events), as well as for exposure, vulnerability, risk/impact and resilience in a multi-hazard context. Many of the papers reviewed in this study (e.g., Li et al., 2021; Lou et al., 2023b; Pal et al., 2023) imply that their results and the use of indicators may be of potential use to stakeholders who are responsible for disaster risk management or climate change adaptation, however, the extent to which stakeholders have been involved in the process of creating and testing indicators to support decision-making in multi-hazard contexts is not clear. Stakeholder engagement varies from consulting with expert groups (e.g., Damian et al., 2023) to interactive co-development (e.g., Fleming et al., 2023). Understanding the priorities, interests, ambitions, and challenges of stakeholders is essential to developing and undertaking effective DRR research (Gill et al., 2021). Of the 194 papers reviewed, however, only 15 studies include stakeholder engagement, of which 6 studies are within the multi-hazard category (i.e., Cremen et al., 2023; Gallina et al., 2020; Hagenlocher et al., 2018; Sekhri et al., 2020; Viavattene et al., 2018; Vitolo et al., 2019) . The remaining 9 studies are either layered single hazard (n=8) or include no specific hazard (n=1). Of the 15 studies that included stakeholder engagement, 14 focused on multi-hazard risk assessment, which requires consideration of socio-economic vulnerabilities and impacts from multi-hazard events. When developing multi-hazard and multi-risk indicators for disaster risk management and climate change adaptation, it is crucial to consider how and where to use multi-hazard information with stakeholders. For example, interactive stakeholder engagement in setting weighting, prioritisation and thresholds plays a critical role, as it guides sensitivity to certain impact areas, such as applying physical drought models to early warning systems for food security (Boult et al., 2022). This approach also enables stakeholders to issue early and timely warnings (Li et al., 2021).

Multi-hazard analysis is inherently interdisciplinary as it involves multiple hazard types that may use different indicators, each requiring distinct analytical methods and datasets. For example, extreme heat and wildfire multi-hazard analysis utilises various datasets and indices: Vitolo et al. (2019) used the Fire Weather Index (FWI) and the Universal Thermal Climate Index (UTCI), while Páscoa et al. (2022) applied the Standardized Precipitation Evapotranspiration Index (SPEI), Number of Hot Days (NHD), and Number of Hot Nights (NHN) in their analysis. Bian et al. (2022) and Qian et al. (2023) developed and refined the Compound Drought Heatwave Magnitude Index (CDHMI) using the Non-Stationary Standardized Precipitation Evapotranspiration Index (NSPEI) and daily maximum temperature (Tmax) to determine the probability of heatwave and drought events. For compounding flood events, Jalili Pirani and Najafi (2022) developed a Compound Hazard Ratio Index to characterise the interactions between different drivers of flooding (e.g., extreme precipitation, river overflows, storm tides) and their effects on return level estimates of compound events. Similarly, Ganguli and Merz (2019) analysed spatio-temporal trends in compound flood events caused by the co-occurrence of fluvial floods and extreme coastal water levels, defining a Compound Hazard Ratio Index that links fluvial discharge with coastal water levels to understand historical trends in compound flooding. Therefore, a collaborative environment across interdisciplinary expertise, with relevant stakeholder engagement, is essential.

## 4.3 Recommendations for improved multi-hazard and multi-risk indicators

Based on these well-established challenges associated with multi-hazard research, and the insights on the use of multi-hazard indicators from our review, we suggest actions that are needed to: (i) advance research and



methodologies that allow robust indicators for multi-hazard contexts, (ii) improve uptake and use of indicators,
and (iii) create an enabling and collaborative environment. These recommendations are intended to support the
ambitions of Sendai Framework for Disaster Risk Reduction and accelerate multi-hazard risk assessment and
management.
*(i) Advancing research into multi-hazard and multi-risk frameworks*
The following recommendations focus on strengthening multi-hazard research aligned with the development, use,
and uptake of multi-hazard, multi-risk indicators:
1.  Advance research into the interrelationships between hazards, including specific coupling
processes.

2.  Advance research into the dynamic components of risk and the interrelationship and overlap
between these components, to accelerate multi-(hazard)-risk assessment.

3.  Advance research into frameworks for multi-hazard risk management that integrate data and
indicators to understand the complexity of multi-(hazard)-risk scenarios, and how to reduce risk in
such contexts.

4.  Develop a robust methodology for capturing, recording, and analysing multi-hazard events and
dynamic interrelationships in multi-hazard environments.

*(ii) Improving uptake and use of indicators*
The following recommendations focus on actions to ensure *multi-hazard and multi-risk* indicators are useful,
useable, and used:
5.  Create guidance for risk management professionals about hazards and their potential interrelated
effects – improving understanding of the multi-hazard and multi-risk concepts, and their
relationship to indicators.

6.  Form partnerships between stakeholders to facilitate cross-disciplinary, cross-sectoral working
arrangements; working with stakeholders to understand indicator requirements and ensure that
developed indicators are relevant and practical for end-users needs.

7.  Create executive summaries and high-level reports that translate technical risk indicator data into
strategic insights for better understanding and decision-making.

8.  Use advanced visualisation tools, such as GIS mapping and interactive dashboards, to present
multi-risk information, indicators, and associated monitoring data in a clear and accessible format.

9.  Extend the use of indicators beyond hazard and risk assessment to establish real-time monitoring
systems and early warning mechanisms that provide up-to-date information on the emergence and
propagation of multi-hazard events.

*(iii) Creating an enabling and collaborative environment*


The following recommendations focus on opportunities to strengthen the collaborative, interdisciplinary environment required for the effective and impactful development, use, and uptake of multi-hazard, multi-risk indicator:

10. Use interdisciplinary expertise for collective knowledge generation and problem-solving to overcome current barriers and challenges in developing effective multi-hazard, multi-risk indicators.

11. Support the development of online open-access collaborative repositories for sharing good practices and data.

12. Develop a multi-hazard, multi-risk indicator library co-developed with stakeholders to provide a centralised data management solution, potentially integrated into the MYRIAD-EU Disaster Risk Gateway open-access, editable wiki (https://disasterriskgateway.net/).

## 5. Conclusions

In this study we systematically reviewed existing multi-hazard and multi-risk indicators and present recommendations for their future development and use. While there is broad use of indicators for risk assessment and management, and for identifying interactions between hazards and warning thresholds, this study finds that there are few studies that explicitly develop indicators for either multi-hazard or multi-risk contexts. The majority of the studies described as multi-hazard or multi-risk were, on inspection, multi-layer single hazard and risk; in other words, these did not include the interactions between hazards. The results also demonstrated a predominance of studies on hazard assessment (88% of publications), and a dominance of meteorological and hydrological hazards, particularly in the context of compounding hazards. Only 20% of the papers included in the review integrated hazard, vulnerability and risk (or impact) – a reflection of the complexity of multi-hazard risk. The methodologies used in the reviewed studies included quantitative, qualitative and mixed methods approaches, with a predominance of mixed methods applied in risk assessment, highlighting the interdisciplinarity and role of methods such as expert judgment in multi-hazard risk assessment. The ongoing challenge related to the selection and use of different multi-hazard risk terminology within the literature was echoed in our findings. Based on the findings of the review, we set out twelve recommendations to progress and enable uptake of indicators, from advancing research into multi-hazard risk frameworks that integrate indicators, to enabling partnerships with stakeholders to ensure the inclusion of their needs and the uptake of indicators in disaster risk management. This review is limited to the peer-reviewed literature; future work could build upon this review through the exploration of grey literature and direct engagement with stakeholders involved in indicator relevant applications of disaster risk reduction (e.g., through interviews).

**Author contribution**

Conceptualization: CW; Data curation: CK; Formal analysis and visualization: MSGA, MA, EN, CK; Methodology and writing – original draft preparation: all authors.

**Competing interests**

At least one of the (co-)authors is a member of the editorial board of Natural Hazards and Earth System Sciences.



**Acknowledgements**
CJW acknowledges support from the NERC Global Partnerships Seedcorn Fund 'EMERGE' project though grant
no. NE/W003775/1. CJW, MSGA, MA, YC and CK were supported by the European Union's Horizon Europe
'Multi-hazard and risk informed system for enhanced local and regional disaster risk management (MEDiate)'
project under grant agreement no. 101074075. MSGA also received support from the Leverhulme Trust through
an Early Career Fellowships [grant reference ECF-2023-074]. RC, JC, MD, LS, and PJW were supported by the
European Union's Horizon 2020 'Multi-hazard and sYstemic framework for enhancing Risk-Informed
mAnagement and Decision-making in the E.U.' (MYRIAD-EU) project under grant agreement no. 101003276.



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
