# Peer review of "Review article: Towards multi-hazard and multi-risk indicators – a review and recommendations for development and implementation"

_Natural Hazards and Earth System Sciences, 2024_

## Author Response (AR1)

**NHESS-2024-178 Review comments**

**RC1**

Dear Editor,

Thank you for giving me the opportunity to review the article "Review article: Towards multi-hazard and multi-risk indicators – a review and recommendations for development and implementation". In the enclosed manuscript, the authors provide a systematic review of recent literature addressing the topic of multi-hazard and multi-risk indicators. The authors clearly identify gaps in the literature and propose a series of recommendations to advance the uptake of indicators in the field of multi-hazard and multi-risk.

**Response:**

Thank you for your detailed and constructive feedback on our manuscript, "Towards Multi-Hazard and Multi-Risk Indicators – A Review and Recommendations for Development and Implementation." We appreciate your thorough review, positive comments and your valuable insights on how to improve the clarity and impact of our work.

**General comment:**

Authors have undertaken a rigorous literature review, bringing together 194 scientific articles addressing a broad range of natural hazards, methods, and disciplines. The objective of the article is to build on the body of literature to provide recommendations on the development of indicators for multi-hazard and multi-risk for disaster risk management and assessment. Objectives are clear and the article is well written. However, I think that the article misses its announced objectives in its current state. The main findings of the article are gaps in the literature. These gaps indeed exist and are well summarized in Section 4. I believe that there is content in the article, and in the literature reviewed, that could shape recommendations that are more impactful. In particular, I see room for improvements on three broad aspects:

**Response:**

Thank you for your general comment. We acknowledge the concern regarding the disconnect between the objectives of the study and the recommendations. To address this, we have refined both the objectives in section 1 and, importantly, recommendations in section 4 so they are now directly informed by the reviewed literature, including highlighting how existing studies contribute to these recommendations and avoid overly generic suggestions that overlap with past studies. Below we address each of the aspects of your comment in turn.

**Disconnection between recommendations (Section 4.3) and results from the literature review:**

Recommendations provided in Section 4.3 do not seem to build on the literature reviewed in their current state. The four recommendations on *Advancing research into multi-hazard and multi-risk frameworks* may result from the fact that only a minority of the reviewed articles address hazard interrelations and impacts. These recommendations are instead very generic, and some have already been made in previous articles (e.g., Ward et al., 2022, Zscheischler et al., 2020). The following eight recommendations also lack connections with results from the literature review.

**Response:**

Building on the previous point, we have extensively revised our recommendations to be more connected to the literature review, providing a clear pathway for their development. Section 4.2 has been refocused and blended with 4.1 (general findings), allowing the revised recommendations in 4.2 (previously 4.3) to more clearly connect with and build upon the results of the review. The number of recommendations has also been reduced to eight to be more succinct and actionable and remove the repetition from previous studies. The revised recommendations in section 4.2 are as follows:

"Based on the insights gained on multi-hazard and multi-risk indicators from this review, and building on previously-established challenges associated with multi-hazard and multi-risk research, we suggest the following eight recommendations that are designed to (i) advance research and methodologies that allow robust indicators for multi-hazard and multi-risk contexts, (ii) improve uptake and use of indicators by providing a clear pathway for their development, and (iii) create and strengthen an enabling and interdisciplinary collaborative environment for their development:

- 1. Indicator development should not solely focus on hazard characteristics but should also integrate risk-based dimensions (e.g., vulnerability, exposure, sensitivity, adaptive capacity) and impacts (physical, economic, environmental), reflecting the complexity of multi-hazards and multi-risks. This development can be extended beyond hazard and risk assessment to establish real-time monitoring systems and early warning mechanisms that provide up-to-date information on the emergence and propagation of multi-hazard events.
- Given the current predominance of indicators for compound multi-hazard events evidenced in the literature, there is a need to develop indicators that capture triggering, amplification, and cascading relationships between hazards to represent the dynamic and interconnected nature of multi-hazard systems.
- 3. Composite indicators designed to capture multi-hazard and multi-risk dimensions should be adaptable to diverse regional contexts, account for socio-economic disparities, and align with the specific priorities of stakeholders, including policymakers, emergency planners, and affected communities.

- 4. Where feasible, mixed-method approaches are essential for developing robust multi-hazards and multi-risks indicators, integrating quantitative data (e.g., historic hazard frequencies, exposure metrics), qualitative insights (e.g., community perceptions), and expert judgement to comprehensively reflect the complexity and interdependencies of risk drivers.
- 5. Multi-hazard and multi-risk indicators should be co-developed through interactive and participatory processes involving relevant stakeholders, ensuring that they are meaningful, practical, and tailored to decision-making needs in disaster risk management and climate adaptation.
- 6. While not specific to indicators, the adoption of clear and consistent terminology in the definition and usage of terms such as 'multi-hazard', 'multi-risk', 'indicator' and 'index' is crucial as ambiguities in terminology currently hinder the comparability and integration differing approaches.
- 7. Indicators should be designed considering the availability, resolution, and quality of underlying datasets, especially where data are scarce or uneven across hazards and/or risks. This can be supported through the use of online open-access collaborative repositories and libraries for sharing good practices and data (e.g., the open-access MYRIAD-EU Disaster Risk Gateway <a href="https://disasterriskgateway.net/">https://disasterriskgateway.net/</a>) together with the use of advanced visualisation tools (e.g., the DRMKC Risk Data Hub <a href="https://drmkc.jrc.ec.europa.eu/risk-data-hub#/atlas">https://drmkc.jrc.ec.europa.eu/risk-data-hub#/atlas</a>).
- 8. Finally, the development of new multi-hazard and multi-risk indicators should align with international frameworks, such as the Sendai Framework for Disaster Risk Reduction, the UN SDGs, and the Early Warnings for All (EW4All) initiative, to ensure these indicators support the measurement, reporting, and achievement of globally recognised targets and contribute effectively o international disaster risk reduction and resilience-building efforts."

We hope these revised recommendations will ensure that the findings of this review contribute more effectively to advancing research in this field.

**Lack of clarity on the content of the reviewed literature:**

The methodology, key words and filters used to compile the 194 article is clearly explained in Section 2. However, some information about the nature of articles collected are not provided, resulting in a difficult interpretation of the result part for the reader. In particular, information about the number of hazards represented, preferably from each category used in the article (meteorological, geophysical ...) would be beneficial. The number of hazards/interrelations considered would also help to understand the state-of-the art in multi-hazard indicators.

**Response:**

We agree that further clarification on the nature of the reviewed literature, particularly the types and distribution of hazards, enhances the interpretation of the results. In response, we have revised the Methodology section to include additional details on the types of hazards identified and the classification approach adopted. Specifically, we now explicitly state that a total of 19 hazard types were extracted from the reviewed

studies and classified into four broad categories based on the UNDRR Hazard Information Profiles (HIPs) (Murray et al., 2021): (1) meteorological and hydrological, (2) geohazards, (3) environmental, and (4) technological. Studies that did not refer to any specific hazard were categorised as 'no hazards'.

To improve clarity, we have also added a new supplementary table (Table S3), which presents the four hazard categories along with the corresponding specific hazard types. This addition aims to provide readers with a clearer understanding of the distribution and representation of hazards within the reviewed literature. Furthermore, we have clarified how hazard interrelationships were addressed in the review. The following revised text has been added to the Methodology section:

"Category 1: Multi-layer single hazard and risk – These studies individually analysed multiple single hazards or risks occurring within a given location, with outcomes presented in an overlaid format. Although often referred to by the authors as multi-hazard or multi-risk, these assessments did not consider interactions between hazards and thus do not meet the definition of multi-hazard as used in this review.

Category 2: Multi-hazard and multi-risk – these studies explicitly addressed interactions between hazards. They were further categorised into two broad classes based on the nature of these interactions: compound; and triggering and amplification relationships. Definitions of these interaction types are also provided in Table 1."

**Presentation of results:**

The alternation of usage raw numbers and percentages in section 3 is confusing. The classification chosen for Figure 3, Table 2 and Table 3 is also confusing, as we are seeing percentages of percentages. The Sankey diagram of Figure 3 is hard to interpret and could be improved. Figure 4 is great. The classification made between compound hazard and cascading hazard indicators in Section 3.3 is also not ideal in my opinion.

Finally, I think that the article has the potential to provide more specific recommendations, building on some existing indicators/indexes (especially for multi-hazard indicators). I therefore suggest major revisions, focusing on Section 3 and Section 4.

**Response:**

Thank you for these valuable suggestions. In response, we have restructured the results section—particularly Section 3—to enhance clarity and coherence.

The original Figure 3, Table 2, and Table 3 have been replaced with a new consolidated Figure 2 (noting that Figure 1 was removed). This updated figure provides a clearer breakdown of both the number and percentage of articles addressing various risk components (hazard, vulnerability, risk, and impact). It also distinguishes between studies focused on interacting multi-hazard events versus multi-layer single hazards, and presents the distribution of methodological approaches (quantitative, qualitative, and mixed methods) used to assess vulnerability, risk, and impact.

To enhance transparency and readability, we have revised the accompanying text throughout the section to clearly specify the subset of articles being discussed (e.g., all articles, hazard-focused articles, etc.). We now report both raw counts (n) and percentages (%) consistently, as recommended.

Additionally, the revised figure caption has been expanded to explain what each percentage represents and how the categories are organised, improving interpretability.

Regarding the classification of compound versus triggering/amplification hazard indicators, we evaluated four primary types of multi-hazard indicators—intensity, frequency, probability, and composite indicators—across all studies. This classification is now clarified in the newly added Table 2. Although both compound and triggering interactions were assessed using the same types of indicators, we acknowledge the need for clearer differentiation. Accordingly, we have revised the text in Section 3.2 (formerly Section 3.3) to better distinguish how indicator types are applied in the context of compound versus triggering multi-hazard events.

**Specific comments:**

1. Line 44-45 p1: "in other words, they did not include the interactions between hazards." Hazard and also risks, no?

**Response:**

We appreciate the reviewer's observation and agree that it is important to clarify that both hazards and risks were considered individually in these studies. Accordingly, we have revised the sentence as follows:

"We found that most studies exploring indicators focused on multi-layer single hazards and risks, where multiple single hazards or risks within a given location were analysed individually and their outcomes presented in an overlaid format."

2. Line 203 p10. This division of "multi-hazard" literature in two groups is relevant and has already been done before (e.g. Tilloy et al., 2019), but it is not obvious to me in the context of the article. Could the authors provide a rational for the division?

**Response:**

The reviewer has identified an important point. Although our review primarily targeted studies that explicitly consider interactions between hazards or risks (i.e., interactive multi-hazard/multi-risk), we found that a considerable number of studies—often self-described as "multi-hazard" or "multi-risk"—in fact analysed multiple single hazards or risks independently, without examining their interrelationships. To account for this and maintain conceptual clarity, we categorised the reviewed studies into two broad groups:

- 1. Multi-layer single hazard and risk, and
- 2. Multi-hazard and multi-risk.

This classification helped us better evaluate how the concept of multi-hazard/multi-risk is operationalised in the literature, especially in relation to the use of indicators. We have clarified this rationale in the revised manuscript. Please refer to the following text:

"Although this review primarily focused on multi-hazard and multi-risk studies that address interactions between hazards or risks, a number of included articles were found to adopt a multi-layer single hazard or risk approach. To distinguish between these different approaches, the 194 reviewed articles were classified into two broad categories:"

3. Line 213-215 p11. It is not clear what is the "vulnerability and impact" category. Does that mean that 88% of all articles deal with hazards, 49% with vulnerability and 35% with impact?

**Response:**

Thank you for this observation. We agree that the original sentence in Section 3.1.1, as well as others in that section, lacked clarity regarding the categorisation of "vulnerability and impact." In the revised manuscript, we have thoroughly rewritten Section 3.1.1 and introduced a new Figure 3 to improve readability and interpretation.

The sentence in question has been clarified as follows:

"This review analysed the use of multi-hazard and multi-risk indicators, focusing on four main categories: hazard, vulnerability, risk and impact, and composite indicators (see Table 2). Figure 2 provides an overview of how the reviewed articles are distributed across the hazard, vulnerability, and risk and impact components. Among the 192 studies included in the review, the components of hazard, vulnerability, and risk and impact were addressed a total of 338 times, as many articles discussed more than one component."

4. Figure 3 p11. The figure is confusing as groups for hazards are "yes" and "no" while they correspond to methods used for vulnerability and impact. Another figure showing how groups of literature overlap would be more informative and connected to the text.

**Response:**

We thank the reviewer for the helpful comment. In response, we have replaced the original Figure 3, Table 2, and Table 3 with a new consolidated Figure 2 (noting that Figure 1 was removed), which provides a clearer overview of how the reviewed articles relate to different risk components: hazard, vulnerability, and risk & impact. The revised figure addresses the confusion caused by the previous "yes"/"no" categorisation and now visually illustrates the overlap between these components, aligning more closely with the discussion in the text. Please refer to the revised Figure 2.

5. Table 3 p12. I do not understand why the "no" column only applies to hazard.

**Response:**

We thank the reviewer for pointing this out. In the revised manuscript, Table 3 has been removed. To improve clarity and address this issue, we have introduced a new Figure 2, which provides a more coherent overview of the articles reviewed in relation to the key risk components: hazard, vulnerability, and risk & impact.

6. Line 259 p13: "The majority of geohazards were associated with multi-layer single hazard interactions (77%)" If it is a multi-layer approach, there are no interactions.

**Response:**

We agree that referring to "interactions" in the context of multi-layer single hazard approaches can be misleading, as these approaches typically involve the overlay of multiple hazard layers without necessarily implying dynamic interactions. In the revised manuscript, we have clarified the wording in Section 3.1.2 to more accurately reflect that these studies adopted a multi-layer methodology without indicating hazard interaction.

7. Line 261 p13. "When it came to multi-hazard interactions, geohazards were 261 almost equally distributed between compound (8%) and triggering and amplification interactions (12%)." That would be great to know which hazards are discussed here.

**Response:**

Thank you for the suggestion. In the revised manuscript, we have now specified the types of hazards involved in compound, triggering, and amplification multi-hazard events. To provide further clarity, we have also included a new supplementary figure (Figure S1), which illustrates the distribution of hazards addressed in multi-hazard studies involving these types of interactions. Please refer to the updated sections in the manuscript, including the following revised text:

"Compound interactions were the second most common, representing 30%(n=149) of all hazards. These interactions involve hazards that occur simultaneously or in close succession. Most compound events stemmed from meteorological and hydrological hazards—particularly drought, extreme temperatures, floods, storms, and extreme precipitation—highlighting their tendency to co-occur and interact in complex ways. A smaller portion of compound hazards originated from geohazards (e.g., earthquakes) and environmental hazards (e.g., wildfires) (Figure S1, Supplementary document)."

"Triggering and amplification interactions accounted for 12% (n=59) of the hazards, where one hazard triggers or amplifies the effects of another. These were predominantly associated with meteorological and hydrological hazards (e.g.,

flooding), followed by geohazards (e.g., earthquakes) and environmental hazards (e.g., wildfires). Finally, 7% (n = 37) of the hazards did not fall into any of the above categories. These were labelled as 'no interaction' cases, either due to limited information or because they did not meet the criteria for multi-layer single hazard, compound, or triggering/amplification relationships (Figure S1, Supplementary document)."

8. Line 267 p14. "We found that approximately 44% of the 194 articles reviewed were categorised as multi-layer single hazard studies, predominantly focusing on meteorological, hydrological, and geo hazards (Fig. 4)." Figure 4 show hazard count and not paper count.

**Response:**

Thank you for pointing this out. After carefully considering this comment along with related feedback from both reviewers, we recognize that the discussion of multi-layer single hazard studies created unnecessary confusion, especially given that the primary focus of our study is on multi-hazard indicators. As a result, we have removed Section 3.2 from the main manuscript and relocated its content to the supplementary document for reference. The specific sentence highlighted in this comment has been deleted from the manuscript. However, key details on multi-layer single hazard studies are still available in the supplementary material for readers who may be interested.

9. Line 287-291 p14. The classification into five hazard groups mentioned implicitly takes into account hazard interrelations. For example: winter weather would include hazards such as extreme snowfall, avalanches, extreme cold, extreme wind... These hazards are interrelated (compound and triggering). Review the sentence accordingly.

**Response:**

Thank you for the observation. As noted in our response to comment #8, the section containing this sentence has been removed from the main manuscript to avoid confusion and maintain focus on multi-hazard indicators. Consequently, the sentence in question is no longer part of the revised manuscript.

10. Line 341 p15. Same as comment h)

**Response:**

Thank you for the observation. As mentioned in our responses to comments #8 and #9, the section containing this sentence has been removed from the main manuscript to reduce confusion and maintain a clear focus on multi-hazard indicators. As a result, the sentence referenced in this comment is no longer included in the revised manuscript.

11. Figure 5. Finally we see the hazards! Although I don't understand how some hazards (e.g., Tsunamis) can be in both categories. The absence of scale make the figure prettier but less interpretable.

**Response:**

We agree that the previous version of Figure 5 (now Figure 4) was less interpretable. In response to this comment, we have replaced the bipartite graph with a new matrix that more clearly illustrates the relationship between primary hazards in multi-hazard sequences and the associated multi-hazard indicators.

Regarding the classification of tsunamis, we acknowledge the confusion. According to the UNDRR's Hazard Information Profiles (HIPs), tsunamis are classified in more than one category. This dual classification has been clarified in the revised manuscript. Please refer to Supplementary Table S3 for further details.

12. Line 374 p17. Drought is also a hydrometeorological hazard, it does not make much sense to separate in from flood here.

**Response:**

We agree that both drought and floods fall under the category of hydrometeorological hazards and separating them in the context mentioned could be misleading. In the revised manuscript, we have retained examples involving both droughts and floods to illustrate the use of compound multi-hazard indicators. To avoid confusion, we have revised the text in Section 3.2.1 to ensure a clearer and more consistent classification of hazards.

13. General p17. The separation of indication for compound and triggering interrelations is not very convincing in this context.

**Response:**

We appreciate the reviewer's observation. In our analysis, we evaluated four main types of multi-hazard indicators—intensity, frequency, probability, and composite indicators—across all studies. This classification is now clarified in the newly added Table 2. While both compound and triggering interactions were assessed using the same indicator types, we acknowledge that the distinction between the two could have been clearer. In response to this comment, we have revised the text in Sections 3.2.1 to more clearly highlight how different indicator types are applied in the context of compound versus triggering multi-hazard events.

14. Line 388 p17. Same as comment h)

**Response:**

Thank you for the comment. As with our response to comment (13), we acknowledge that the distinction between compound and triggering multi-hazard interactions required further clarification. To address this, we have revised the text in Section 3.2.2 to more clearly illustrate how the four main types of multi-hazard indicators—intensity, frequency, probability, and composite indicators—are applied in studies focusing on triggering interactions. This classification is also now clearly outlined in the revised Table 2.

15. Line 484 p22. This first recommendation was already made in Kappes et al. (2012) more than a decade ago.

**Response:**

As above, we have revised the recommendations significantly following your earlier general comments, which has removed this particular recommendation.

16. Line 487-491 p22. Why "interrelationship" is suddenly being used in these lines (more occurrences here than in the previous 21 pages)?

**Response:**

Thank you for pointing this out. The term has been used earlier in the paper, but we appreciate that its usages increased here. This has been addressed together with the revision to the recommendations more broadly (detailed above).

17. Line 502 p22. You mean Section 4.2?

**Response:**

This has been removed.

18. Line 586 p24. This is not new. It is even a requirement for EU funded projects.

**Response:**

We agree that executive summaries and high-level reports are indeed not new, however, our point here was to highlight that

19. Line 588 p24. Be aware that similar tools already exist. Examples: https://drmkc.jrc.ec.europa.eu/risk-data-hub#/atlas

**Response:**

Thank you for this suggestion. This has been incorporated.

20. General conclusion: Please state that you identified gaps in the literature

**Response:**

This has now been included in the conclusions thank you.

21. Line 618-620 p25. It is not clear for me that recommendations are based on the literature review.

**Response:**

Thank you. As above, we have refined both the objectives and recommendations so they are directly informed by the reviewed literature, including highlighting how existing studies contribute to these recommendations and avoid overly generic suggestions and overlap with past studies.

**RC2**

**Dear Editor,**

Thank you for the invitation to review the article "Towards multi-hazard and multi-risk indicators – a review and recommendations for development and implementation" which I enjoyed reading and believe can make a valuable contribution to the literature with minor to medium level revisions. The paper outlines a robust literature review of articles that mention indicators, which are subsequently categorised by approach to multi-hazards and component of risk they address.

The introduction is strong and the recommendations for future work are clear, and I am confident the review has been undertaken rigorously. Most of my comments about the paper closely align with the comments of reviewer 1. I feel the three key areas where the paper could be enhanced are:

**Response:**

Thank you for your constructive feedback on our manuscript. We are grateful for your detailed comments and agree that addressing these points will enhance the paper. Below we outline how we have revised the manuscript following your general and minor comments.

**General comments:**

**1. Presentation of the results**

Similar to reviewer 1, the presentation of results in Section 3.1, Figure 3, Table 2 and 3 could be presented more clearly. It is not always clear when you are talking about a subset of the whole corpus of papers, or a subset of a subset, so I suggest using more words to make clear which subset you are referring to and using both raw numbers and percentages e.g., 49% of 'ABC' articles (n= XX) where 'ABC' might represent 'all' or 'hazards-focussed' etc.. I like the idea of Figure 3, but I found it somewhat overcomplicates quite a simple paragraph, and some of the percentages in-text do not clearly match the diagram (e.g., 29% of articles mentioned on line 219 seems to equate to 34% of articles in the Sankey diagram – perhaps I am misinterpreting something here, but this emphasises that the diagram is not easy to follow!). Similarly, both Table 2 and 3 would benefit from a more detailed figure caption which indicates what the percentages represent (i.e., % of all articles or % of a subset – I started to get a bit lost on whether rows/columns should add up to 100%) and further labelling of columns/rows to indicate what these represent.

**Response:**

Thank you for these valuable suggestions. In response, we have restructured the results section—particularly Section 3.1—to enhance clarity and coherence.

We have replaced the original Figure 3, Table 2, and Table 3 with a new consolidated Figure 2 (noting that Figure 1 has been removed), which presents a clearer breakdown of

the number and percentage of articles addressing different risk components (e.g., hazard, vulnerability, risk, and impact). The figure also distinguishes between articles focusing on interacting multi-hazard events and those analysing multi-layer single hazards. Furthermore, it illustrates the distribution of methodologies (quantitative, qualitative, and mixed methods) used to assess vulnerability, risk, and impact.

To improve transparency throughout the manuscript, we have revised the accompanying text to explicitly state which subset of the dataset is being discussed (e.g., all articles, hazard-focused articles, etc.). We now consistently report both the raw numbers (n) and percentages (%) for each subset, as recommended.

We have also ensured that the revised figure and accompanying caption clearly explain what each percentage represents and how categories are organized, thereby improving readability and interpretability. We appreciate your feedback in helping us present these findings more clearly.

**2. Bringing the indicators upfront and further synthesis**

I believe the strongest contribution to the literature can make is specific guidance about indicators, rather than further commentary on the complexity and/or absence of genuine multi-hazard indicators. At present, indicators are largely spoken in quite an 'abstract' sense until Section 3.2, and then this section is rather descriptive of individual papers rather than giving a strong sense of synthesis. Firstly, I would like to see more examples of specific indicators mentioned in Section 1 (beyond the box about the Sendai framework). I encourage the authors to consider if Table 4 could be expanded upon to really give a sense of the outputs of the systematic review, and brought right to the front of the results section in a brief overview (or if it is large, placed into supplementary material and a summary given in-text). Then, in Section 3.2, most paragraphs could do with a couple of sentences at the end to synthesise the findings of that paragraph (at present, most of the paragraphs start with a signposting sentence and then describe the methods/outputs of papers one by one).

**Response:**

Thank you for this excellent observation. This is the same broad conclusion we reached after we submitted in hindsight. As such, the indicators have now been brought forward in the paper much earlier, with Box being completely rewritten to focus on indicators now with examples. The results in section 3 have undergone similar changes and have been extensively reworked to bring out more details from the results, including an expansion of Table 3 (was Table 4).

**3. Connection between the results and the discussion**

Although the recommendations for further action are good, much like reviewer 1, it is sometimes hard to see how these are evidenced by the results section (e.g., the mention of dynamic risk is very limited up until the discussion). I believe that reviewer

1's suggestions are an appropriate way forward for improving the link between the two sections.

**Response:**

Thank you for this comment. Aligned with the comments from RC1 (and our responses to them), we have revised the recommendations in the discussion. We have reinforced the links between findings and recommendations, ensuring that each suggestion is now supported by the literature review results.

**Minor revisions:**

Line 83 'For example, a precipitation indicator...' – it would be helpful to add an example indicator here.

**Response:**

To improve clarity and provide specific examples, we have revised the sentence as follows:

"For example, a precipitation indicator such as the Standardized Precipitation Index (SPI) may be used to represent meteorological drought (AghaKouchak et al., 2023), while cumulative rainfall thresholds or intense rainfall events (e.g., daily precipitation exceeding the 90th percentile) may be used as indicators of flood occurrence (Papagiannaki et al., 2022)."

Line 136 Typo says 'Section 33'

**Response:**

Corrected.

Paragraph starting line 148 Add a sentence indicating the process for arriving at this set of search terms to indicate the robustness of the method (i.e., that you are not missing specific terms).

**Response:**

To address this comment, we have included the following sentence in the revised manuscript:

"To ensure methodological rigor and minimise the omission of relevant studies, keywords were carefully selected to maximize coverage of pertinent literature while limiting the retrieval of irrelevant results, following best practices for systematic reviews (Pullin and Stewart, 2006)."

Line 172 'Relevance was primarily assessed.' consider adding the word 'manually' here – just to help distinguish that some parts of your methods are done in a semi-automated way using R.

**Response:**

Thank you. We have added the word 'manually' in this sentence.

Line 173 It is unclear why only geo and hydrometeorological hazards are considered here – this has not been mentioned up until this point.

**Response:**

In the revised manuscript, we have excluded 'geo and hydrometeorological origin' from this sentence. Please refer to the following sentence:

"Relevance was primarily assessed manually based on the use of multi-hazard and multi-risk indicators in evaluating natural hazards across diverse research domains."

Paragraph starting line 206. This paragraph needs expanding upon to explain how you arrived at these five classes and what these classes are (could go into a table).

**Response:**

Thank you. In response, we have expanded the paragraph to clarify how indicator categories were selected. In addition, Table 2 has been added to the manuscript to describe each category with representative examples.

Line 310 It would be useful to give a broad overview (one or two sentences) with some signposting of what the results section is going to cover.

**Response:**

Thank you for this helpful suggestion. Line 310 was part of Section 3.2 in the original manuscript. After carefully reviewing this comment along with related feedback from both reviewers, we recognized that the inclusion of multi-layer single hazard studies created unnecessary confusion—particularly since the main focus of our study is on multi-hazard indicators.

Therefore, we have removed Section 3.2 from the main manuscript and relocated its content to the supplementary materials for reference. The specific sentence noted in this comment has been deleted. However, key details on multi-layer single hazard studies remain accessible in the supplementary document for readers interested in that aspect of the analysis.

Table 3 Figure caption – In text you refer to methods rather than indicators, so the word methods should appear in the figure caption.

**Response:**

Thank you for your observation. As noted in our response to the first comment, the original Table 3 has been removed from the revised manuscript. In its place, we have reorganized the presentation of results and updated the relevant figures. Additionally, we have revised the captions of all figures to enhance clarity and ensure they accurately reflect the content—particularly with respect to whether they describe methods, indicators, or other elements.

Figure 4. This is a nice figure but the colour scheme makes it very hard to visualise the smallest categories on both screen and in print. Explore using a diverging colour scheme.

**Response:**

Thank you. We have revised Figure 4 (now Figure 3) by applying a diverging colour scheme to improve visual clarity, particularly for the smallest categories.

Sentence starting line 278 and ending on line 280 should be supported with a citation.

**Response:**

Thank you for your suggestion. The sentence beginning on line 278 was part of Section 3.2 in the original manuscript. However, as previously mentioned, we have removed this section to maintain the study's focus on interactive multi-hazard events. Key details related to multi-layer single hazard studies remain available in the supplementary document for readers who are interested in that aspect of the analysis.

Figure 5A. There appear to be more hazards than there are circles, which means the labels are misaligned with the circles. This gets particularly hard to read around storm surge and soil salinity. If there is a hazard category that does not correspond to a circle, consider adding a cross or a lighter circle to make this clearer.

**Response:**

Thank you for the helpful observation. We agree that the previous version of Figure 5a (now Figure 4), particularly the bipartite graph, posed interpretability challenges due to label misalignment and unclear representation of certain hazards. In response to this and related comments, we have replaced the bipartite graph with a redesigned matrix. The new figure more clearly illustrates the relationships between primary hazards in multi-hazard sequences and their associated multi-hazard indicators, improving both clarity and readability.

Table 4. I like this table and would like to see expansion of this. I would encourage the authors to experiment with fleshing this table out a little bit more and/or adding a visualisation (as mentioned above). A few thoughts for the authors to explore:

- At present, some indicators are listed in detail such as coastal flooding, and then others are spoken about quite broadly such as exposure sensitivity and resilience. The text for other indicators implies that the description in the table is finite, but actually there are many ways of measuring – for example landslides -% of area with steep slopes is rather an approximate indicator compared to others in the literature. I think what is needed is a bit more consistency in how the indicators are outlined.
- At present, it is not really clear which citation relates to which indicator for example, only 3 references are given for 7 indicators for hazard. This makes it difficult for the reader to look in further detail as it is not clear which citation relates to which indicator. It might result in a lot of repetition, but could you explore adding the citations in line with the indicators?
- At present, it is hard to get a sense of what 'sets' of indicators are appropriate or commonly used – I wonder if the authors could explore a network diagram or Sankey diagram to show this.

**Response:**

We thank the reviewer for the thoughtful and constructive feedback. In response, we have revised Table 3 (was Table 4 in the original manuscript) to enhance both clarity and consistency. Specifically, we have:

- Expanded the table content to provide a more detailed and uniform description of the indicators used across studies, ensuring that each category is presented with consistent depth.
- Clarified citation alignment by linking each indicator directly with its corresponding reference(s), allowing readers to more easily trace the source of specific indicators for further investigation.
- Organised indicators under the broader classification scheme introduced in Table 2, thereby improving coherence across sections.

While we appreciate the suggestion to include a visualisation such as a Sankey or network diagram, we note that only 18 studies in our review employed multi-risk indicators. Given this limited sample size, a visualisation would not sufficiently capture meaningful patterns or relationships. Therefore, we have chosen to prioritise clarity and depth in the extended table format, which we believe more effectively conveys the diversity and application of multi-risk indicators across the reviewed literature.

These revisions are reflected in Section 3.2.3 of the manuscript and in the updated Table 3.

---

## Referee Report (RR1)

**Dear Dr White et al.,**

Thank you for taking into account my previous comments and giving me a chance to review again your manuscript "Review article: Towards multi-hazard and multi-risk indicators – a review and recommendations for development and implementation".

First, I want to stress the excellent work from the authors in addressing Reviewer 2 and my comments. The new classification of the literature reads much better and the recommendations are now well aligned with the content of the review. While the article is very much improved, I have some concerns about Section 3.2. and in particular the parts addressing hazard indicators. Beside this point, I provide a few minor comments.

**Comment related to Section 3.2:**

Sections 3.2.1 and 3.2.2 discuss the usage of multi-hazard indicators for multiple types of hazard interactions. I find these sections very lengthy, and sometimes confusing, as they mostly enumerate indicators used in previous studies addressing several hazards. The length of these sections contrasts with one of the main conclusion of the article: "there are few studies that explicitly develop indicators for multi-hazards". Notably, several indicators enumerated are single hazard indicators (combined to create multi-hazard indicators) and two paragraphs are dedicated to one hazard interaction: hot-dry. The other paragraph of Section 3.2.1 is dedicated to compound flooding. I believe that this highlights the lack of compound hazard indicators, and that three paragraphs may not be necessary to make that point. Section 3.2.2 introduces more single hazard indicators, while it is no clear what indicators are discussed in the two last paragraphs (l. 351-l. 362). On the other hand, the Global Delta risk Indicator (Section 3.3) would maybe deserve more discussions. I would therefore suggest to shorten Sections 3.2.1 and 3.2.2 and focus on existing multi-hazard indicators.

**Specific comments:**

- 1) Box1, paragraph 2, 1.1 environmental variables?
- 2) L.93 p5: The development of multi-hazard and multi-risk indicators for disaster risk assessment and management has, however, not kept pace with the development of multi-hazard DRR approaches and the use of indicators more generally. Repetition of sentence in Box1.
- 3) L. 116-118 p5: Sentence feels like a rewording of the clearer sentence in L.229-230 p7
- 4) L.141 p7: Third time you mention that you used a systematic review.
- 5) L. 166 p8: Is it the number of sources obtained after post 2015 reduction?
- 6) L. 175-177 p9: Specify here that you detail the exclusion criteria just below.
- 7) L. 263-265 p13: refer to Table S3 here
- 8) L. 278 p14: most common what?
- 9) L.281 p14: why complex?
- 10) L. 303 p16: Figure number missing
- 11) L.303 p16: Unclear what are primary hazards here, are we only looking at triggering and amplification interrelation?
- 12) Table 3: Costal exposure index is an example of vulnerability indicator?
- 13) L.485-488 p23: Three categories for 8 recommendations, either you group the recommendations or your remove the numbering from categories.

---

## Referee Report (RR2)

I thank the authors for their detailed reply and commend their substantial revisions to the paper which make it much clearer. The new figures and tables really help to break things down. This paper will be an excellent addition to the literature.

I have a few further minor comments - these are mainly suggestions to add in a few additional sentences to help with clarity, plus a few minor typos. The main point that should be addressed is that in the results section, some of the paragraphs read a bit like a 'telephone directory' approach to a literature review - basically describing a set of papers sorted into themes - several paragraphs here could do with a final sentence or two to tell us what that set of papers collectively tell us. This will not require any re-analysis or significant time.

**Comments:**

It is unclear why articles that discussed no hazards and no interaction mechanisms were still included within the scope of the study. A sentence in section 3.1.2 or the methods to give an indication of what types of article this encompasses would be helpful.

Line 314 you set up the argument that the 'design and application of indicators varied' and then go on to describe a range of indicators. This paragraph could do with a final sentence to come back to the argument about the design/application based on what you have described - e.g., why does this matter or what do we now know from your synthesis of these studies?

Generally throughout section 3.2.2, each paragraph could do with a final sentence of summary or synthesis. At present, the structure of the paragraphs is an opening signposting sentence indicating the theme of the indicators you are looking at, followed by a descriptive list of a range of papers. It doesn't come across very clearly what these examples collectively show, or what the insight is from your analysis.

Bullet point starting on line 420 feels like it slightly contradicts your findings that 51% of the articles account for interactions between hazards (I was surprised it was this high!). This also appears in conclusions on line 528. Consider adding another sentence around line 420 to be more specific about what you think is missing from these studies.

Bullet points starting on line 420, it would be helpful to refer back to either sections of your findings or figures to help tie this together (just by section/figure number in brackets). I say this as the fourth and fifth bullet points are interesting but I don't think came across particularly clearly in your results. You might want to think about this with regards to my comment above that some of the paragraphs could do with a final sentence to synthesise/reflect and more explicitly make statements related to terminology and stakeholder engagement.

Writing on lines 461 - 466 feels more like results (and then if moved to results, possibly would answer my point above about the key finding of lack of stakeholder engagement not being that clear in the results section).

Sentence on line 478 about EU funded projects - would be nice to reference a few of these projects or calls

Recommendation 3 in section 4.2 did not come through very clearly in the results or discussion. Consider how you could bring this through more clearly in the results section (or if it is already there, perhaps refer to the section it appears in).

Line 133 - typo 'section 33'
Line 238 need a space after studies, line 301 need a space after study
Line 303 figure number is missing

---

## Author Response (AR2)

**NHESS-2024-178 R2 Review comments**

**RC1**

Thank you for taking into account my previous comments and giving me a chance to review again your manuscript "Review article: Towards multi-hazard and multi-risk indicators – a review and recommendations for development and implementation".

First, I want to stress the excellent work from the authors in addressing Reviewer 2 and my comments. The new classification of the literature reads much better and the recommendations are now well aligned with the content of the review. While the article is very much improved, I have some concerns about Section 3.2. and in particular the parts addressing hazard indicators. Beside this point, I provide a few minor comments.

**Response:**

Thank you for taking the time to re-review our manuscript, "Towards Multi-Hazard and Multi-Risk Indicators – A Review and Recommendations for Development and Implementation" and for your positive comments about the R1 revision. We note the reviewer's additional minor comments, especially with regards to section 3.2, which we address in turn below.

**Comment related to Section 3.2:**

Sections 3.2.1 and 3.2.2 discuss the usage of multi-hazard indicators for multiple types of hazard interactions. I find these sections very lengthy, and sometimes confusing, as they mostly enumerate indicators used in previous studies addressing several hazards. The length of these sections contrasts with one of the main conclusion of the article: "there are few studies that explicitly develop indicators for multi-hazards". Notably, several indicators enumerated are single hazard indicators (combined to create multi-hazard indicators) and two paragraphs are dedicated to one hazard interaction: hot-dry. The other paragraph of Section 3.2.1 is dedicated to compound flooding. I believe that this highlights the lack of compound hazard indicators, and that three paragraphs may not be necessary to make that point. Section 3.2.2 introduces more single hazard indicators, while it is no[t] clear what indicators are discussed in the two last paragraphs (l. 351-l. 362). On the other hand, the Global Delta risk Indicator (Section 3.3) would maybe deserve more discussions. I would therefore suggest to shorten Sections 3.2.1 and 3.2.2 and focus on existing multi-hazard indicators.

**Response:**

We appreciate the reviewer's insightful comment. In response, we have substantially revised Sections 3.2.1 and 3.2.2 to reduce length and improve clarity. These sections now focus more directly on categorising the types of multi-hazard indicators used in the literature—specifically, composite, frequency, intensity, and probability-based indicators—as they relate to compound and triggering/amplification multi-hazard events. The revised text places greater emphasis on the methodological characteristics and limitations of existing multi-hazard indicators, rather than enumerating singlehazard metrics or detailing specific hazard combinations (e.g., hot-dry). Please refer to the revised tracked changes in Sections 3.2.1 and 3.2.2.

We also agree with the reviewer that the Global Delta Risk Index (GDRI) deserves further discussion. In response, we expanded its treatment in Section 3.2.3 to highlight its conceptual depth, multidimensional nature, and applicability across delta regions. The revised paragraph reads as follows:

"Composite risk indicators were also widely adopted across multi-risk studies. A notable example is the Global Delta Risk Index (GDRI), which provides a comprehensive framework for assessing risks in vulnerable delta regions exposed to multiple hazards such as cyclones, floods, storm surges, and droughts (Hagenlocher et al., 2018; Gallina et al., 2016; Depietri et al., 2018; Zhang et al., 2023). The GDRI is designed to evaluate social—ecological systems holistically, capturing the interplay between environmental hazards and human wellbeing. It enables spatial analysis of risk components (e.g., exposure, ecological and social susceptibility, and the robustness of ecological systems as coping mechanisms) at the sub-delta administrative scale, supporting both crossdelta and inter-delta comparisons (Cremin et al., 2023). Another application of composite indicators was observed in Gotangco and Josol (2022), which developed the Physical Service Index (PSI) framework to evaluate the combined effects of urban development, flooding hazards, and chronic deprivation at the regional scale in Manila, Philippines."

**Specific comments (1-13)**

Box1, paragraph 2, l.1 environmental variables?

**Response:**

We have changed "environmental parameters" to "environmental variables" on the first line of paragraph 1 in Box 1.

L.93 p5: The development of multi-hazard and multi-risk indicators for disaster risk assessment and management has, however, not kept pace with the development of multi-hazard DRR approaches and the use of indicators more generally. Repetition of sentence in Box1.

**Response:**

We thank the reviewer for pointing out the repetition. To address this, we have rephrased the sentence in line 93 to better reflect the motivation for this study and to avoid redundancy with Box 1. The revised sentence now reads:

"However, the development of indicators that specifically address multi-hazard and multi-risk scenarios has lagged behind the broader advancement of multi-hazard DRR strategies and the general growth of risk indicators".

L. 116-118 p5: Sentence feels like a rewording of the clearer sentence in L.229-230 p7

**Response:**

We appreciate the reviewer's observation. Upon review, we believe the intended reference was to lines 134–136 on page 7 in the previous version of the manuscript. To address the redundancy, we have removed the sentence previously located at lines 116–118 on page 5 in the revised manuscript.

L.141 p7: Third time you mention that you used a systematic review.

**Response:**

Thank you for pointing out this redundancy. As the Introduction already states that a systematic review was conducted, we have revised the opening sentence of the Methods section to avoid repetition. The revised sentence now reads:

"This study employed a structured approach to identify peer-reviewed literature that either use indicators or analyse their applications in multi-hazard and multi-risk contexts. The process was guided by the Preferred Reporting Items for Systematic reviews and Meta-Analyses (PRISMA) protocol (Page et al., 2021)."

L. 166 p8: Is it the number of sources obtained after post 2015 reduction?

**Response:**

Yes, the duplicate removal process was applied to the dataset after excluding pre-2015 articles, as well as non-English and inaccessible records. To clarify this sequence, we have revised the paragraph accordingly. The updated text now reads:

"The initial search returned 1,468 articles that met the search criteria. A publication date filter was then applied to include only studies published from 2015 onwards in alignment with the release of the Sendai Framework for Disaster Risk Reduction and its emphasis on multi-hazard approaches. After excluding the pre-2015 publications, non-English articles, and inaccessible records, 1,140 articles remained. A duplicate removal process, conducted using the R programming language, identified and eliminated 515 duplicates from this set. Figure 1 provides a flowchart detailing the screening process, including the number of articles at each stage of the review."

L. 175-177 p9: Specify here that you detail the exclusion criteria just below.

**Response:**

Thank you for the suggestion. To clarify that the exclusion criteria are detailed immediately after, we have revised the sentence as follows:

"Relevance was primarily assessed manually based on the use of multi-hazard and multi-risk indicators in evaluating natural hazards across diverse research domains, with detailed exclusion criteria outlined below."

L. 263-265 p13: refer to Table S3 here

Response:

Done.

L. 278 p14: most common what?

**Response:**

Thank you for the comment. We have clarified that the sentence refers to hazard interactions. The revised sentence now reads:

"Compound interactions were the second most common hazard interactions, representing 30% (n=149) of all hazards."

L.281 p14: why complex?

**Response:**

Thank you for the comment. To clarify the use of the term "complex," we revised the sentence to better explain the nature of these interactions. The updated sentence now reads:

"Most compound events stemmed from meteorological and hydrological hazards particularly drought, extreme temperatures, floods, storms, and extreme precipitation highlighting their tendency to co-occur and interact across different temporal and spatial scales, which contributes to their complexity."

L. 303 p16: Figure number missing

**Response:**

Thank you for pointing this out. We have corrected the text by including the appropriate figure reference—Figure 4a—at the relevant location in the revised manuscript.

L.303 p16: Unclear what are primary hazards here, are we only looking at triggering and amplification interrelation?

**Response:**

Thank you for this comment. We believe the confusion likely arose due to the previously missing figure reference, as noted in the reviewer's earlier comment. In this context, the term "primary hazards" refers to those shown in Figure 4a.

Table 3: Costal exposure index is an example of vulnerability indicator?

**Response:**

Thank you for the question. In Table 3, the category "Exposure/Vulnerability indicators" encompasses both exposure and vulnerability measures. The Coastal Exposure Index is listed as an example under the "Exposure Index" sub-type, reflecting its role in quantifying exposure rather than vulnerability alone.

L.485-488 p23: Three categories for 8 recommendations, either you group the recommendations or your remove the numbering from categories.

**Response:**

Thank you for your suggestion. The 8 recommendations, which had previously been reworked from the original submission, don't neatly fit into the 3 categories. Indeed, we don't view these as categories but rather high-level themes that the 8 recommendations address collectively. To avoid confusion, we have removed the (i) to (iii) numbering, but have left the recommendations numbered 1-8, which we prefer to keep.

**RC2**

I thank the authors for their detailed reply and commend their substantial revisions to the paper which make it much clearer. The new figures and tables really help to break things down. This paper will be an excellent addition to the literature.

I have a few further minor comments - these are mainly suggestions to add in a few additional sentences to help with clarity, plus a few minor typos. The main point that should be addressed is that in the results section, some of the paragraphs read a bit like a 'telephone directory' approach to a literature review - basically describing a set of papers sorted into themes - several paragraphs here could do with a final sentence or two to tell us what that set of papers collectively tell us. This will not require any reanalysis or significant time.

**Response:**

Thank you very much for taking the time to re-review our manuscript, "Towards Multi-Hazard and Multi-Risk Indicators – A Review and Recommendations for Development and Implementation" and for your positive comments about our R1 revision. We note the reviewer's additional comments, which we address below.

**Comments**

It is unclear why articles that discussed no hazards and no interaction mechanisms were still included within the scope of the study. A sentence in section 3.1.2 or the methods to give an indication of what types of article this encompasses would be helpful.

**Response:**

We thank the reviewer for highlighting this important point. To clarify, articles that did not discuss any hazards or interaction mechanisms were not included in the analysis of multi-hazard and multi-risk indicators. However, in Section 3.1.2, we explain why some articles categorised as having "no hazards" still appear in Figure 3. Specifically, the following sentence provides clarification:

"The review noted that although some articles discussed hazards in general, no specific hazard types according to the UNDRR's HIPs classification were addressed."

Additionally, we have clarified that articles without explicit interaction mechanisms (i.e., "no interactions") were excluded from the analysis of multi-hazard and multi-risk indicators. This is noted in Section 3.2, where we state:

"This study only considered interactive multi-hazard events to identify multi-hazard and multi-risk indicators, as discussed in Section 3.2."

Line 314 you set up the argument that the 'design and application of indicators varied' and then go on to describe a range of indicators. This paragraph could do with a final sentence to come back to the argument about the design/application based on what

you have described - e.g., why does this matter or what do we now know from your synthesis of these studies?

**Response:**

Thank you for the insightful suggestion. Following the revision of Sections 3.2.1 and 3.2.2 in response to RC1's review comments above, which now focus on key categories of multi-hazard indicators (composite, frequency, intensity, and probability) related to compound, triggering, and amplification events, we have added concluding sentences that explicitly synthesize the implications of the diversity in indicator design and application. This clarifies why these differences matter and what insights can be drawn from the range of studies reviewed.

Generally throughout section 3.2.2, each paragraph could do with a final sentence of summary or synthesis. At present, the structure of the paragraphs is an opening signposting sentence indicating the theme of the indicators you are looking at, followed by a descriptive list of a range of papers. It doesn't come across very clearly what these examples collectively show, or what the insight is from your analysis.

**Response:**

As part of the restructuring mentioned above, the revised Section 3.2.2 now incorporates summary sentences at the end of each paragraph to clearly articulate the collective insights derived from the examples. This improved structure better highlights current practices and gaps in the development and use of multi-hazard indicators, grouped under the categories of composite indicators, hazard frequency, hazard intensity, and hazard probability.

Bullet point starting on line 420 feels like it slightly contradicts your findings that 51% of the articles account for interactions between hazards (I was surprised it was this high!). This also appears in conclusions on line 528. Consider adding another sentence around line 420 to be more specific about what you think is missing from these studies.

**Response:**

Thank you for this comment. To clarify, the 51% number relates to papers that assessed multi-hazards (n = 89 of 174 in total) and included a reference to or a mention of indicators (see Methods for a more complete description). However, this number doesn't necessarily mean that 51% of these papers developed or used multi-hazard indicators – rather it means that indicators are mentioned, referenced and, in some cases used, in the context of multi-hazards. This was an important step in the review methodology that enabled us to then get to the few papers that did indeed develop and/or use indicators in this context, which we show progressively from Figures 2 to 4. We therefore elected to not highlight the 51% number as a key finding or takeaway from the paper. Instead, we focus on highlighting the relatively few examples of multi-hazard (and later multi-risk) indicators, which is the focus of this first key finding bullet point. With this added clarity, we believe this bullet point is therefore not contradictory and as

such we haven't made any further changes. However, noting the following reviewer's comment, we have added references back to the text for each key finding so the reader can now more easily link back to the relevant section that has led to it. In addition, we have amended the first line of Section 3.1.1 to reiterate the methodology used in the review to avoid confusion later on, as follows:

"This review analysed papers that assessed multi-hazards and/or multi-risks and included reference to or mention of indicators...".

Bullet points starting on line 420, it would be helpful to refer back to either sections of your findings or figures to help tie this together (just by section/figure number in brackets). I say this as the fourth and fifth bullet points are interesting but I don't think came across particularly clearly in your results. You might want to think about this with regards to my comment above that some of the paragraphs could do with a final sentence to synthesise/reflect and more explicitly make statements related to terminology and stakeholder engagement.

**Response:**

Thank you for this comment. We have added in references to the relevant sections / figures / tables at the end of each key finding. We have also taken the opportunity to slightly refine the text in a couple of these bullet points for clarity.

Writing on lines 461 - 466 feels more like results (and then if moved to results, possibly would answer my point above about the key finding of lack of stakeholder engagement not being that clear in the results section).

**Response:**

Thank you – this is a good observation and suggestion. We have moved this text to the second-to-last paragraph of Section 3.2.3, which now provides this information as a result, enabling the findings and recommendations to follow on from this.

Sentence on line 478 about EU funded projects - would be nice to reference a few of these projects or calls.

**Response:**

Thank you for this suggestion. We have added in a couple of examples.

Recommendation 3 in section 4.2 did not come through very clearly in the results or discussion. Consider how you could bring this through more clearly in the results section (or if it is already there, perhaps refer to the section it appears in).

**Response:**

Thank you. We agree that this did not come across that clearly in the results. However, the revision of Section 3.2.3 in response to one of RC1's review comments above now include a more detailed discussion of composite indicators with reference to regional contexts and socio-economic settings. However, we note that the final part of this recommendation is perhaps unsubstantiated in the results, hence we have removed the final "including policymakers, emergency planners, and affected communities" wording.

| Line 133 - typo 'section 33'.                                           |
|-------------------------------------------------------------------------|
| Response:                                                               |
| Corrected.                                                              |
|                                                                         |
| Line 238 need a space after studies, line 301 need a space after study. |
| Response:                                                               |
| Corrected.                                                              |
|                                                                         |
| Line 303 figure number is missing.                                      |
| Response:                                                               |
| Corrected by including Figure 4a.                                       |